Report

EMBO
reports

# Tudor-based proteomic strategy pan-specifically enriches and identifies protein arginine methylation

Lingzi Lu [ID] [1,5], Ting Li [1,5], Rou Zhang [ID] [1,5], Yutong Wang [1], Xiaoping Ye [1], Yixin Luo [1], Lingyu Sun [2], Liang Qi [1], Zilu Ye [ID] [3✉], Yang Mao [ID] [1,4✉] & Yanqiu Yuan [ID] [1✉]

## Abstract

Protein arginine methylation is an important post-translational modification (PTM) in eukaryotes, regulating a variety of biological processes. Proteomic profiling of arginine methylation has advanced our understanding of its roles in biology and disease. However, pan-specific enrichment of methylarginine-containing peptides remains challenging. Herein we report a molecular affinity strategy based on the Tudor domain of SMN, a naturally occurring methylarginine reader protein, for comprehensive proteomic profiling of cellular arginine methylation. We demonstrate that the Tudor domain-based approach exhibits broad specificity for proteins harboring mono- or di-methylated arginines, encompassing both RGG/RG-rich and non-RG motifs, facilitating the discovery of novel methylation sites. Using this strategy, we identify asymmetric dimethylarginine (aDMA) in protein eIF3D, an essential component of the eukaryotic translation initiation complex. Biochemical analyses reveal that aDMA modification at R99 of eIF3D plays a regulatory role in protein translation initiation. Our findings establish a generally applicable approach for proteomic profiling of arginine methylation and unveil its novel regulatory role for this modification in eukaryotic protein translation.

**Keywords** Arginine Methylation; eIF3D; Molecular Affinity Enrichment; Proteomics
**Subject Categories** Methods & Resources; Post-translational Modifications & Proteolysis

## Introduction

Arginine methylation, the addition of one or two methyl groups to the ω-nitrogen of arginine residues, is a conserved protein post-translational modification (PTM) among mammalian cells (Bedford and Clarke, 2009; Bedford and Richard, 2005; Blanc and Richard, 2017). Much like lysine methylation, arginine methylation has been extensively studied as a crucial epigenetic mark for transcriptional regulation (Guccione and Richard, 2019; Wu et al, 2021). Beyond histones, arginine methylation is increasingly recognized in non-histone proteins, indicating its wide-spread role as a PTM that regulates diverse biological processes, such as DNA damage and repair, mRNA splicing, chromatin remodeling and protein–protein interactions (Al-Hamashi et al, 2020; Biggar and Li, 2015; Lorton and Shechter, 2019; Wei et al, 2014). The development of novel enrichment methodologies for comprehensive mapping of protein arginine methylation holds the potential to uncover previously unknown regulatory functions.

However, arginine methylation does not substantially alter the positive charge of the modified residue, and the increase in hydrophobicity conferred by a single methyl group is marginal (Blanc and Richard, 2017). Consequently, the specific enrichment of methylated arginine-containing peptides for proteomic analysis is challenging. Over the past decade, considerable effort has been directed toward developing methods to selectively distinguish between modified and unmodified arginines for effective enrichment (Guo et al, 2014; Li et al, 2021; Ma et al, 2017; Wang et al, 2024; Wang et al, 2016; Wang et al, 2022; Wang et al, 2011). Chromatography methods employing either high pH strong cation exchange (Hartel et al, 2019; Li et al, 2021; Wang et al, 2016), high-pH reversed-phase chromatography (Wang et al, 2019), or hydrophilic interaction liquid chromatography (Ma et al, 2017; Uhlmann et al, 2012) have been developed. However, these methods tend to enrich clustered arginine methylation regions. The development of antibody-based enrichment strategies, which utilize antibodies recognizing monomethyl arginine (MMA), asymmetric dimethylarginine (aDMA) or symmetric dimethylarginine (sDMA), integrated with mass spectrometry-based global proteomic analysis (Guo et al, 2014; Hartel et al, 2019; Maron et al, 2021; Mulvaney et al, 2021), has demonstrated high specificity and significantly facilitated the identification of arginine-methylated proteins. Nevertheless, these antibodies often exhibit sequence preferences due to the limited diversity of antigens used in their generation. For example, commercially available antibodies developed for sDMA preferentially recognize methylarginines within RGG/RG motifs (Lu et al, 2024; Musiani et al, 2019; Radzisheuskaya et al, 2019). Recently, a steric effect-based chemical

[1]State Key Laboratory of Anti-Infective Drug Discovery and Development, School of Pharmaceutical Sciences, Sun Yat-sen University, Guangzhou, China. [2]Guangdong Institute for Drug Control, Guangzhou, China. [3]State Key Laboratory of Common Mechanism Research for Major Diseases, Suzhou Institute of Systems Medicine, Chinese Academy of Medical Sciences & Peking Union Medical College, Suzhou, China. [4]Guangdong Provincial Key Laboratory of Drug Non-Clinical Evaluation and Research, Guangzhou, China. [5]These authors contributed equally: Lingzi Lu, Ting Li, Rou Zhang. ✉E-mail: yzl@ism.pumc.edu.cn; maoyang3@mail.sysu.edu.cn; yuanyq8@mail.sysu.edu.cn

enrichment method coupled with boronate-affinity chromatography has been developed to efficiently enrich dimethylated arginines at the proteome level (Wang et al, 2022). This method offers an alternative strategy for arginine methylation profiling, although it may necessitate specialized chemical expertise. Consequently, an easily applicable, "pan-specific" enrichment method capable of profiling arginine methylation independent of flanking sequence context remains a crucial need.

An affinity reagent engineered from the MBT domain repeats of L3MBTL1, which specifically bind mono- and dimethylated lysine, was developed as a general tool for detecting and global proteomic profiling of methyllysine (Moore et al, 2013). To address a similar challenge in methylarginine profiling, we explored the potential of a naturally occurring methylarginine binding Tudor domain as a molecular affinity reagent. The Tudor domain of SMN protein is known to bind both sDMA and aDMA, as well as MMA with a lower affinity (Tripsianes et al, 2011). Importantly, it can bind methylarginines in the form of free amino acids, suggesting the binding is relatively independent on primary sequences flanking the methylation sites. Therefore, the SMN Tudor domain may have the combination of pan-specificity toward methylarginines and high methyl selectivity required for global proteomic studies.

Here, employing the molecular affinity strategy based on the SMN Tudor domain, coupled with an optimized mass spectrometry method for arginine methylation profiling developed in our group (Lu et al, 2024), we demonstrate efficient enrichment of methylarginine-containing peptides from cell lysate. This approach enables the identification of both dimethylarginine (DMA) and monomethylarginine (MMA) sites within both RGG/RG-rich and non-RG motifs, highlighting the broad specificity of our strategy. This strategy complements existing methylarginine profiling methods and may facilitate the discovery of previously unrecognized functions of arginine methylation. In this study, we identify a novel aDMA site in the non-RG motifs of eIF3D and reveal a new regulatory role for arginine methylation in mammalian protein translation initiation. Compared to previous enrichment methods, our affinity purification strategy requires a relatively small amount of sample input, suggesting its potential utility in profiling physiologically relevant samples.

## Results and discussion

### Tudor-based molecular affinity strategy could enrich methylarginine-containing peptides with high efficiency

Structural analysis of the SMN Tudor domain with a dimethylarginine-containing peptide bound in its aromatic cage also suggests that the residues flanking the methylation mark do not directly contribute to their binding (Fig. 1A). This encouraged us to build a molecular affinity approach based on SMN Tudor for pan-specific enrichment of methylarginines in proteomic analysis. To develop such an approach, we first measured the binding affinity between SMN Tudor and three types of methylarginine, MMA, sDMA and aDMA, using isotherm titration calorimetry (ITC). We show that SMN Tudor binds to sDMA, aDMA, and to a lesser extent, MMA with a $K_D$ at 0.359, 0.860 and 1.8 mM, respectively (Table 1 and Fig. EV1). As expected, it does not bind to non-methylated arginine up to 5 mM and a single mutation in the aromatic cage (Y127L, W102F or N132E) abolishes the recognition

(Appendix Fig. S1), confirming its binding specificity. To confirm that flanking residues do not significantly affect methylarginine-Tudor domain interactions in our system, we synthesized heptapeptides derived from the Sm D1 protein sequence, each containing either an aDMA (SmD1-aDMA) or an sDMA modification (SmD1-sDMA) and performed ITC experiments with the SMN Tudor protein. As demonstrated in Table 1, the binding affinities of SMN Tudor for SmD1-aDMA and SmD1-sDMA were comparable to those observed with free methylarginine amino acids. This similarity suggests that flanking sequences exert minimal influence on binding, thereby reinforcing the mechanistic principle that SMN Tudor domain specificity is predominantly driven by methylarginine recognition rather than by contextual sequence features. Encouraged by the results, we then constructed an expression plasmid encoding a fusion protein consisting of an SMN Tudor domain, a C-terminal His$_6$ tag and an N-terminal Halo-tag, which could be immobilized onto the Halo-link resin through a covalent reaction (Fig. 1A). The fusion protein was expressed in E. coli, purified through Ni-NTA and immobilized onto Halo-link resin. The maximum loading capacity was estimated to be 360 µg protein per 100 µL resin measured by SDS-PAGE with Coomassie blue staining (Appendix Fig. S2), which was used as a reference for all future preparations.

We first assessed if this Tudor affinity resin could enrich methylarginine peptides by applying the tryptic digest of HeLa cell lysate (Fig. 1B). Following two washes with the binding buffer and subsequent washes with water, the bound peptides were eluted with 0.15% TFA and cleaned up using C18 StageTip before being subjected to LC-MS analysis. A hybrid fragmentation method, ETciD, was used for optimal methylarginine profiling as previously reported (Lu et al, 2024). Shotgun proteomics could identify between 586 and 742 methylarginine peptides including both monomethyl and dimethyl modifications with an enrichment factor between 50.0% and 67.7% in each replicate experiment (Fig. 1C and Dataset EV1). This result suggests that the Tudor affinity resin can specifically enrich methylarginine peptides with good efficiency. Combining three datasets, we have identified 841 methylarginine sites using the Tudor affinity strategy (Fig. 1D), with an overlap of 62.5% between the three replicates (Fig. 1E). The intensity of precursor ions also correlates well among the three datasets (Appendix Fig. S3), confirming the high reproducibility of the molecular affinity-based enrichment method. The number of DMA sites identified by this approach is comparable to those reported using aDMA- and sDMA-specific antibodies (Guo et al, 2014; Lim et al, 2020; Lu et al, 2024; Ma et al, 2023a). It's worth noting that only 360 µg of trypsin-digested peptide sample was used in each replicate experiment, a significantly lower quantity compared to what is typically required for antibody enrichment (Guo et al, 2014; Hartel et al, 2019). Additionally, no further fractionation was involved, making this method a flexible and cost-effective alternative for the proteomic profiling of arginine methylation. The reduced sample requirement renders this method potentially applicable for the analysis of samples of physiological relevance and of limited availability.

### Global proteomic profiling by molecular affinity strategy identifies novel arginine methylation sites

A detailed analysis of identified methylarginine peptides revealed that these peptides carry one to four methylarginines (Fig. 2A).

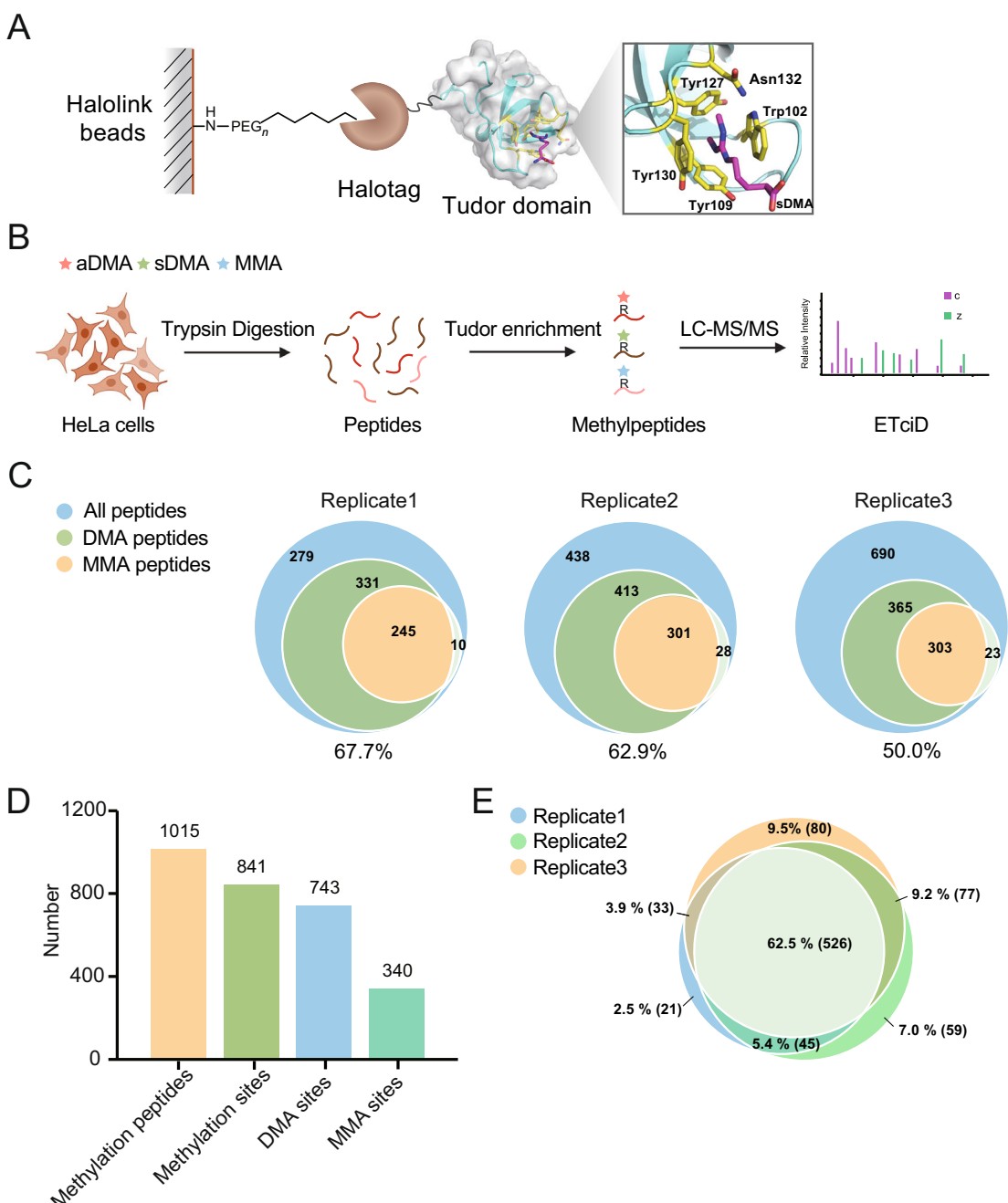

**Figure 1. Molecular affinity strategy for proteomic profiling of arginine methylation.**

(A) Schematic diagram depicting the Tudor affinity resin. SMN Tudor domain shown in surface representation (light gray) is immobilized onto the Halo-link agarose resin via an N-terminal Halo-tag. The box on the right illustrates the recognition of sDMA by the aromatic cage of SMN Tudor domain. (B) Workflow for the molecular affinity strategy for arginine methylation profiling. (C) The enrichment efficiencies of molecular affinity strategy in three biological replicate experiments. (D) The number of methylation peptides and different methylarginine sites (monomethyl or dimethyl) identified from three biological replicates. (E) The reproducibility of molecular affinity strategy by comparing identified methylarginine sites in three biological replicates.

Importantly, ~10.5% of the identified peptides contain only one methylation mark, confirming this Tudor domain-based molecular affinity strategy can identify isolated methylarginine sites which are not easily enriched by chromatography-based methods. It is also worth noting that the majority of the peptides carry at least one dimethyl moiety, and a small fraction contains MMA only (Fig. 2B).

This is consistent with ITC findings that the SMN Tudor domain prefers to bind to dimethylarginine and binds weakly to MMA. Sequence analysis around these sites revealed 68.7% of RGG/RG repeats and 31.3% of non-RGG/RG repeats (Fig. 2C), indicating the broad specificity of the strategy. In addition, the adjacent amino acids to the methylated arginines in non-RG peptides showed

Table 1. Thermodynamic parameters for binding between SMN Tudor protein and various methylated arginines.

| Ligand | $K_D$ (mM) | ΔH (kcal/mol) | −TΔS (kcal/mol) | ΔG (kcal/mol) |
|--------|-----------|---------------|------------------|----------------|
| sDMA | 0.359 ± 0.045 | −3.71 ± 0.209 | −0.99 | −4.70 |
| aDMA | 0.860 ± 0.034 | −6.05 ± 0.140 | 1.87 | −4.18 |
| MMA | 1.800 ± 0.322 | −1.92 ± 0.242 | −1.83 | −3.75 |
| Arginine | >5 mM | n.c. | n.c. | n.c. |
| SmD1-sDMA | 0.375 ± 0.023 | −14.50 ± 0.503 | 9.82 | −4.67 |
| SmD1-aDMA | 0.834 ± 0.046 | −8.02 ± 0.313 | 3.82 | −4.20 |
| SmD1 | >2 mM | n.c. | n.c. | n.c. |

A fixed stoichiometry (N) of 1 and a 1:1 binding mode was employed for all data fitting.
*n.c.* not calculable.

minimal sequence specificity, only exhibiting a slightly increased frequency of proline at the +1 position. This observation further underscores the broad applicability and specificity of the enrichment methodology. Molecular function analysis of the identified proteins showed a significant enrichment in RNA/DNA binding (Fig. 2D), which is consistent with the known functions in which arginine methylation was involved.

We next focused on DMA sites, of which there has been more systematic prior analysis, and further compared the identified dimethylation sites in our datasets with those in the PhosphoSitePlus database, which is a comprehensive, manually curated resource for post-translational modifications (Hornbeck et al, 2015). As shown in Fig. 2E, of the 743 identified dimethylation sites, 391 sites (52.6%) were previously reported in the database, and 352 sites are newly discovered, which yields 46 new proteins with arginine dimethylation. The results suggest that our strategy is complementary to the existing methods and can be used to identify unknown dimethylation sites with unknown biological functions. A further comparison between the DMA sites from our study and those reported in Uniprot showed that our method could identify both known aDMA and sDMA sites (Appendix Fig. S4), reflecting the specific recognition of both DMA forms by the SMN Tudor protein as demonstrated by the ITC results. Nonetheless, further research is needed to distinguish between sDMA and aDMA identified in the study, as these two types of methylation share the same molecular weight and cannot be efficiently differentiated based on mass spectrometry spectra alone.

Given the isobaric nature of methylation to several amino acid substitutions, false positive identification can occur in LC-MS analysis (Hart-Smith et al, 2016). To measure the confidence in protein methylation identification in our enrichment, we employed the heavy methyl SILAC (hmSILAC) method, which directly labels post-translationally modified methyl groups by culturing cells in heavy methionine ([$^{13}CD_3$] methionine) (Ong et al, 2004). After heavy methyl groups were fully incorporated into protein methylation sites, peptides were digested and enriched by Tudor affinity resin and subjected to LC-MS analysis as described above. As a result, 84 methylation peptides were validated using the hmSILAC method, with a validation rate of 49.1% (Fig. 2F), which

is comparable to previously reported profiling strategies (Lu et al, 2024; Massignani et al, 2019). These results further confirmed that our molecular affinity strategy is effective for the global profiling of protein arginine methylation.

## eIF3D is dimethylated in the RNA-binding region

Dimethylation at R97, R99 and R103 in the eukaryotic initiation factor 3D protein is among the newly validated modification sites by our molecular affinity strategy (Figs. EV2A and 3A). The peptide does not contain RGG/RG repeats, nor does it have PGM (proline-, glycine- or methionine-rich) motifs typified by PRMT4 substrates (Cheng et al, 2007; Shishkova et al, 2017). However, methylation in this stretch of peptide also comes in tandem, as found in many RNA-binding proteins, indicating a potential functional significance as clustered methylation signifies potential regulatory functions in protein–protein or protein–RNA interactions (Bedford and Richard, 2005; Guccione and Richard, 2019). eIF3D is a conserved member of the eukaryotic translational initiation complex 3 and plays an important role in global protein synthesis by coordinating 48S translation initiation complex (Brito Querido et al, 2020). The identified methylation sites are in the RNA binding region (86–118) of eIF3D, preceding the cap-binding domain (Fig. 3B). Although this RNA binding region of eIF3D is not shown in the EM structure of human 48S translation initiation complex (Brito Querido et al, 2020), potentially due to its high flexibility as seen with highly methylated RGG/RG motifs, early crosslinking experiments (Asano et al, 1997) and its relative location in 48S suggest that it may coordinate the binding of mRNA and 18S rRNA (Fig. 3C). In addition, multiple sequence alignment showed that arginine residues in this region of eIF3D are conserved in eukaryotes from Drosophila to humans (Fig. 3D). Therefore, we hypothesize that arginine methylation in this region may have a conserved regulatory role in eukaryotic protein translation initiation.

To investigate the role of eIF3D methylation, we first need to reveal the chemical nature of the modifications, since sDMA and aDMA are isobaric and difficult to be distinguished by MS. By manual examination of the MS$^2$ spectrum of the eIF3D dimethylated peptide (96–106: NR$_{me2}$MR $_{me2}$FAQR $_{me2}$NLR) (Fig. 3A), we found fragment ions corresponding to neutral losses of dimethylamine (DMA, NH(CH$_3$)$_2$, 45.06 Da), dimethylcarbodiimide (DMC) and dimethylguanidine (DMG), unique to aDMA modification. Furthermore, the absence of a diagnostic ion corresponding to monomethylamine neutral loss in the MS$^2$ spectrum, a signature of sDMA fragmentation, further substantiates the chemical nature of these DMA sites. To confirm that both R99 and R103 are modified with aDMA, we examined the MS$^2$ spectrum of short peptides containing only R99 or R103 (Fig. EV2B). Both spectra exhibited neutral losses of dimethylamine, indicative of aDMA modifications.

To provide further evidence for the type of methylation in eIF3D protein, we overexpressed full-length Flag-eIF3D protein in HEK293T cells, immunoprecipitated with Flag antibody, and performed western blot analysis with pan-specific aDMA and sDMA antibody. However, neither antibody detected the presence of dimethylation modification in the immunoprecipitated samples (Appendix Fig. S5). The result was not unexpected because the commercially available sDMA and aDMA antibodies have sequence

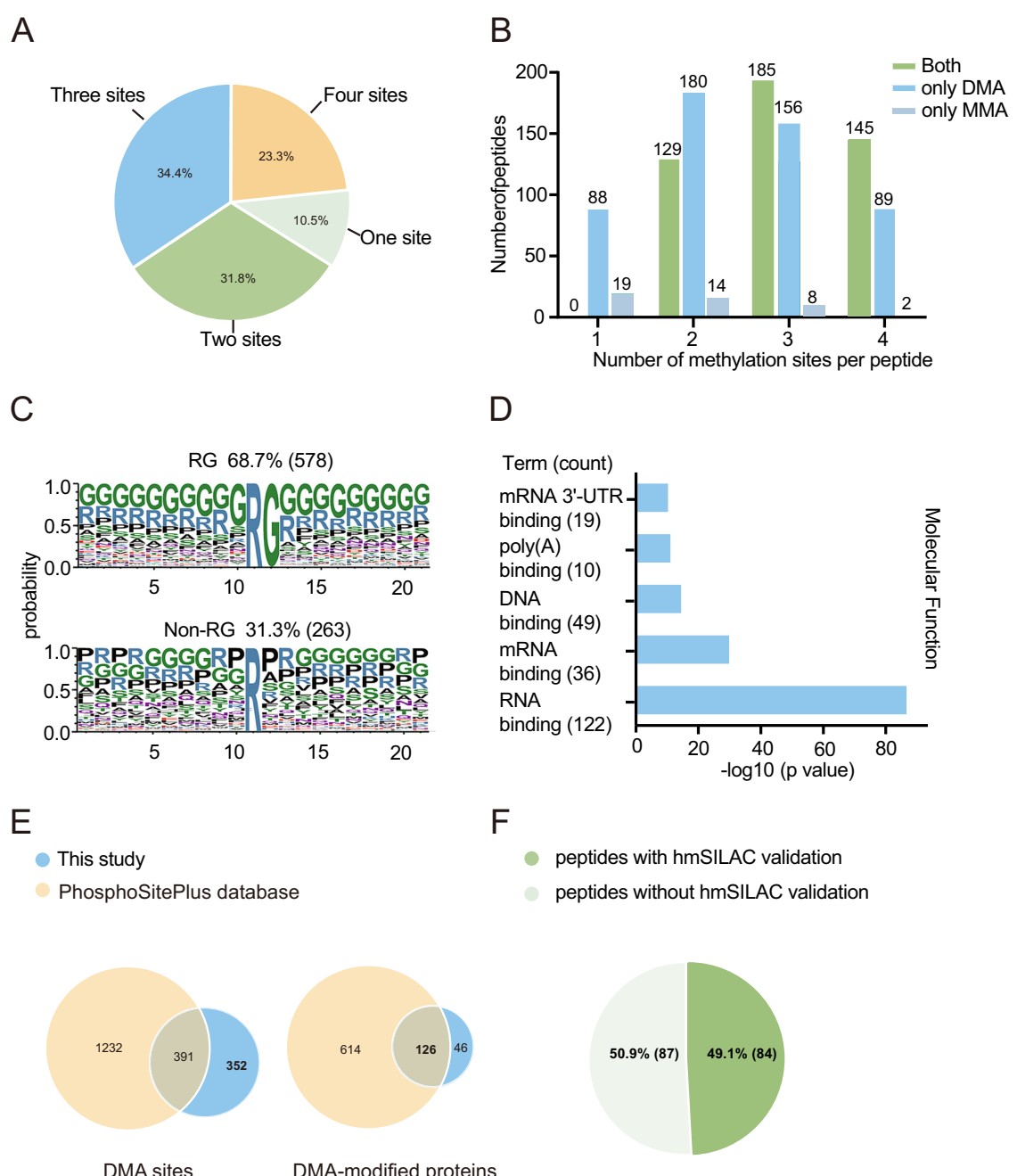

**Figure 2. Detailed analysis of methylation sites identified by molecular affinity approach.**

(A) Percentage of peptides containing 1, 2, 3, or 4 methylation sites. (B) Number of peptides containing 1, 2, 3, or 4 methylation sites with different modification types (dimethyl or monomethyl arginine only or both). (C) Motif analysis of methylarginine sites by molecular affinity approach. (D) Molecular function enrichment of methylarginine proteins by molecular affinity approach using David software ($P < 0.001$). $P$-values were calculated using the hypergeometric distribution and adjusted by the Benjamini and Honchberg method. (E) Overlaps of DMA sites and DMA-modified proteins between this study and PhosphoSitePlus database. (F) Percentage of methylarginine peptides validated by hmSILAC experiments.

preferences and may not be able to detect modifications in this region.

To address this problem, we first carried out in vitro enzymatic assays using purified PRMTs, including PRMT1, 2, 3, 4, 6, 8 and PRMT5:MEP50 complex, a synthetic peptide from eIF3D (86–106) as the substrate and monitored the reactions by MALDI-TOF-MS.

We found that Type I PRMTs including PRMT1, PRMT4 and PRMT6 can catalyze multiple methylation events, adding up to 4 methyl groups on the peptide in vitro, while other PRMTs including Type I PRMT2, PRMT3, PRMT8 and Type II PRMT5:MEP50 protein complex cannot (Fig. 3E), supporting that the methylation modifications on eIF3D are aDMAs.

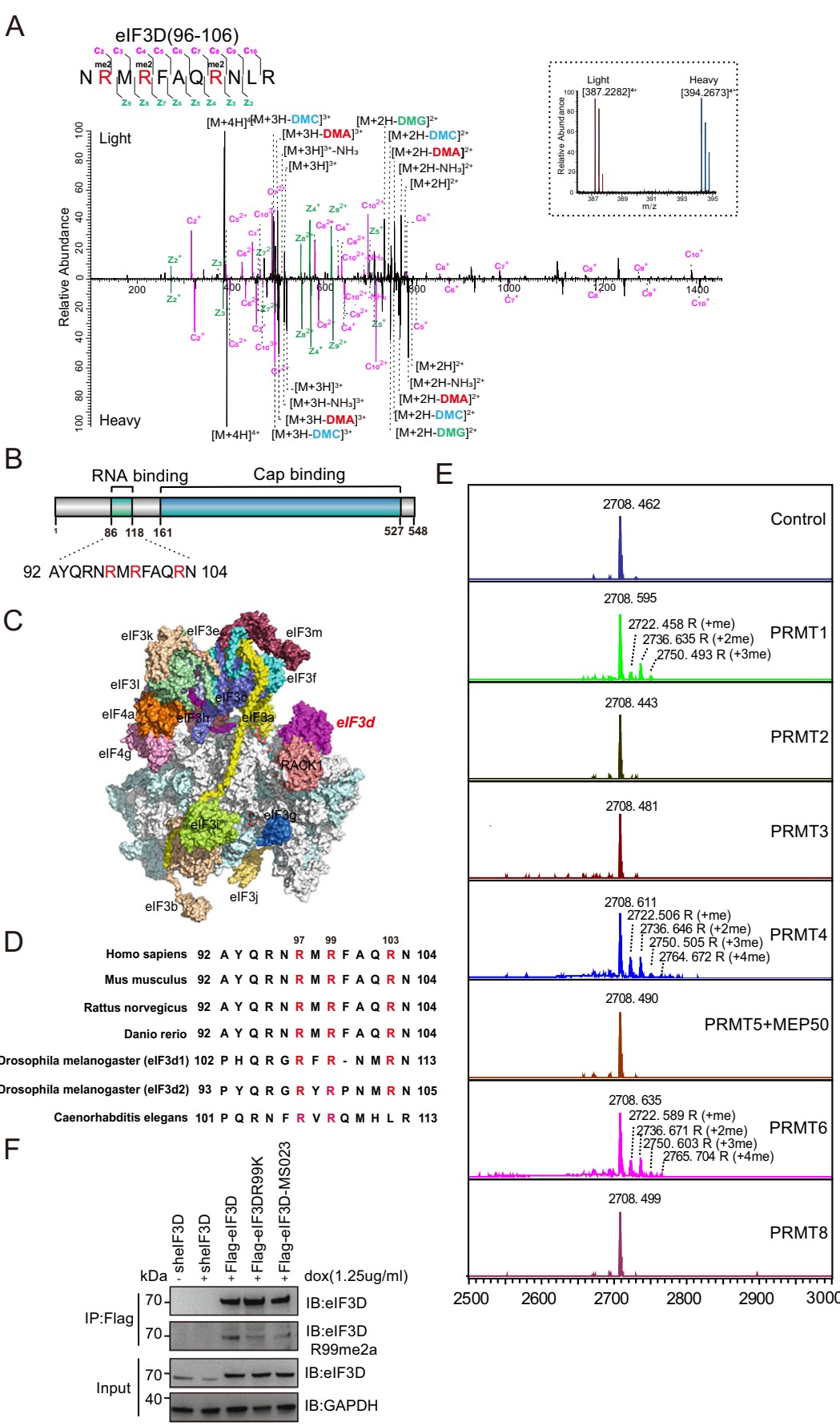

**Figure 3.   eIF3D is dimethylated at R97, R99 and R103.**

(A) MS/MS spectrum of the eIF3D peptide NR$_{me2}$MR$_{me2}$FAQR$_{me2}$NLR (96–106) from the hmSILAC experiment. MS1 of the corresponding hmSILAC doublet is shown in the inset. The spectrum was manually annotated with potential neutral losses. DMA, dimethylamine; DMC, dimethylcarbodiimide; DMG, dimethylguanidine. The abbreviations include their corresponding ions. (B) Schematic illustration of eIF3D protein domain structure. The arginine methylation sites in RNA binding region are shown in red and labeled. (C) A structure model of human 48S complex (PDB 7QP7). All eIFs, mRNA and rRNAs are shown in surface representation. eIF3D is shown in magenta; other eIFs are shown and labeled in their corresponding colors; 18s rRNA and 40S protein S are shown in palecyan and gray, respectively; mRNA is shown in cyan and red. (D) Conservation of the primary sequence in the RNA binding region of eIF3D among eukaryotes. R97, R99 and R103 are marked in red. (E) MALDI-TOF-MS analysis of in vitro methylation reactions with eIF3D peptide (86–106) and PRMT1, 2, 3, 4, 6, 8 and PRMT5:MEP50 complex. (F) Immunoblot analysis of the eIF3D R99 aDMA modification with R99K mutation and after MS023 treatment. Flag-tag wild-type eIF3D or R99K mutant were stably expressed in HEK293T cells with endogenous eIF3D knocked down by doxycycline inducible shRNA. Inhibitor treatment was performed with HEK293T cells overexpressing wild-type eIF3D with 1 µM MS023 for 72 h. Source data are available online for this figure.

We then custom-made a site-specific antibody against the eIF3D R99 aDMA modification (anti-R99me2a), which was selected based on its specific recognition of R99me2a-containing peptide, but not the unmodified peptide (Appendix Fig. S6). The specificity of the antibody was also shown toward wild-type but not the R99K mutant of full-length Flag-eIF3D protein which was overexpressed in HEK293T cells and immunoprecipitated with Flag antibody (Fig. 3F). The R99K mutation was designed to simulate a methylation-free condition at R99 of eIF3D. In addition, cellular treatment with the generic type I PRMT inhibitor MS023 caused a significant decrease in the signal intensity against eIF3D using anti-R99me2a antibody (Fig. 3F), confirming that the modification on R99 of eIF3D is aDMA and is catalyzed by type I PRMT.

In summary, we identified and confirmed the presence of dimethylation in the RNA-binding region of eIF3D through proteomic analysis by molecular affinity enrichment and hmSILAC. In vitro enzymatic assay and site-specific antibody confirmed that R99 of eIF3D is asymmetrically dimethylated by type I PRMTs.

## eIF3D physically interacts with and is modified by PRMT1

In the in vitro enzymatic assay, PRMT1, 4 or 6 can catalyze the methylation of the eIF3D peptide. To further understand which PRMT is responsible for the cellular level of R99 methylation, we knocked down (KD) PRMT1 using shRNAs and knocked out (KO) PRMT4 and/or PRMT6, respectively, in HEK293T (Appendix Fig. S7). Western blot analysis using anti-R99me2a antibody showed that PRMT1 knockdown resulted in a significant reduction in aDMA modification at R99 of eIF3D (Fig. 4A), but complete knockout of PRMT4 or PRMT6 or both led to no obvious change (Fig. 4B–D), suggesting that PRMT1 is responsible for R99me2a modification, at least in HEK293T cell. The functional role of PRMT1 in eIF3D R99 methylation was crossed validated by Immunoprecipitation Mass Spectrometry (IP-MS). eIF3D over-expressed in HEK293T cells with and without PRMT1 knockdown was immunoprecipitated and submitted to LC-MS analysis. Quantification of mass spectrometry peaks corresponding to the dimethylated R99-containing peptide (MR$_{me2}$FAQR) revealed an approximately 40% reduction in peak intensity following PRMT1 knockdown (Fig. EV3). This observation corroborates previous western blot analysis, indicating PRMT1 as the enzyme catalyzing R99 dimethylation. However, the persistence of residual dimethylation suggests the involvement of redundant enzymatic activities targeting R99 in the absence of PRMT1. To further investigate this hypothesis, we pharmacologically inhibited type I PRMTs using MS023, followed by IP-MS analysis of eIF3D. A more substantial

reduction of 65% in the peak intensity of dimethylated R99 was observed, indicating potential contributions from other PRMTs to R99 dimethylation. Nevertheless, the specific identity of these contributing PRMTs remains to be elucidated.

Furthermore, the immunoprecipitation experiment in HEK293T cells overexpressing Flag-PRMT1 and Myc-eIF3D showed direct interactions between eIF3D and PRMT1 (Fig. 4E), suggesting that PRMT1 could physically interact with eIF3D in the cell and be responsible for its methylation modifications.

Taken together, our in vitro enzymatic reaction results and genetic KD/KO experiments using site-specific antibody anti-R99me2a and mass spectrometry, suggest that PRMT1 catalyzes the cellular asymmetric dimethylation of eIF3D.

## aDMA modification at R99 of eIF3D is involved in the translation initiation complex and regulates cap-dependent translation initiation

To demonstrate the functional importance of R99 dimethylation in eIF3D, we first confirmed the incorporation of R99 dimethylated eIF3D in the translation initiation complex. We purified the eIF3 complex by overexpressing Flag-Myc-tagged eIF3L followed by immunoprecipitation using anti-Flag antibody. Western blot and Coomassie stain confirmed the presence of eIF3D in the complex (Figs. 5A and EV4A). MS analysis of the eluent identified dimethylated R97, R99, and R103 of eIF3D in this purified eIF3 complex (Fig. EV4B), suggesting potential role of eIF3D arginine methylation in protein translation initiation. Next, we performed sucrose gradient sedimentation and fractionation of HeLa cell lysates for the isolation of the 48S complex following GMP-PNP treatment, which is the functional unit for the initiation of canonical eukaryotic protein translation. Western blot analysis confirmed the presence of eIF3D and rpS19 in the fractions containing 48S (Fig. 5B) and MS analysis of the trypsin-digested sample again identified peptides containing dimethylated R97, R99 and R103 (Fig. 5B), thereby further support the role of eIF3D arginine methylation in protein translation.

eIF3D is known to be an essential component of the eIF3 initiation complex in the canonical eIF4E-dependent translation initiation (Brito Querido et al, 2020; Merrick and Pavitt, 2018). It is also reported to function as the m$^7$G cap-binding protein in the eIF4E-independent alternative translation initiation for mRNAs carrying specific 5′UTR structures such as that of c-JUN (Lee et al, 2016; Ma et al, 2023b). Therefore, to quantitatively measure the impact of eIF3D R99 methylation on translation initiation, we constructed a luciferase reporter assay that monitors either

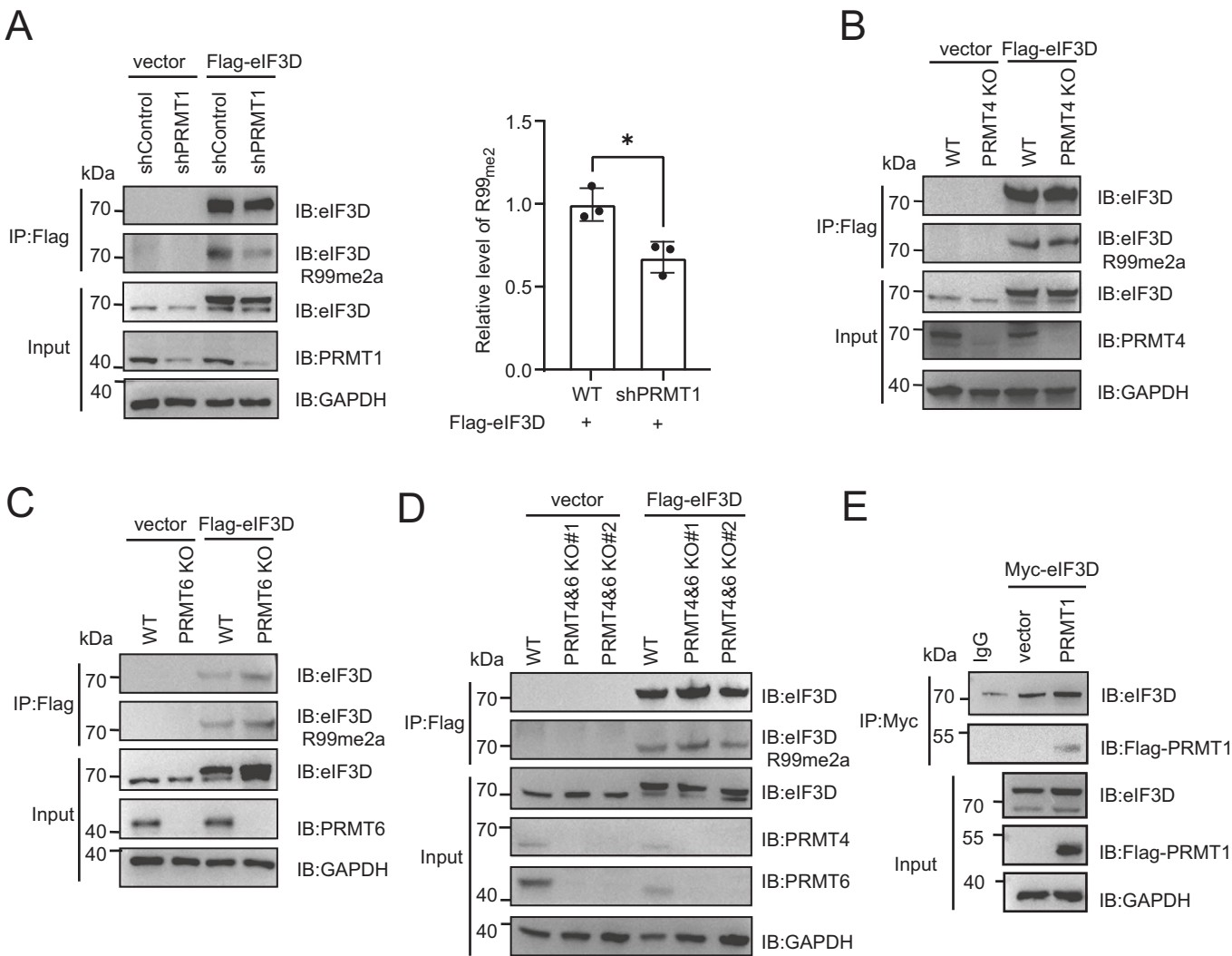

**Figure 4. eIF3D physically interacts with and is modified by PRMT1.**

(A–D) Immunoblot analysis of the eIF3D R99 aDMA modification in wild-type HEK293T cells and HEK293T cells with PRMT1 knockdown (**A**), PRMT4 knockout (**B**), PRMT6 knockout (**C**), or PRMT4 and PRMT6 double knockout (**D**). For (**A**), statistic data are presented as mean ± SD; *$P = 0.0161$ by unpaired t test; $n = 3$. (**E**) Immunoprecipitation analysis of the interaction between eIF3D and PRMT1 in HEK293T cells. Cells were co-transfected with Myc-eIF3D and pcDNA3.1 empty vector or Flag-PRMT1 plasmid. The Myc-tagged protein was immunoprecipitated by anti-Myc antibody and analyzed by western blot. Source data are available online for this figure.

canonical eIF4E-dependent or the alternative cap-dependent translation in the cell. This was achieved by cellular transfection of m⁷G capped mRNAs carrying either a canonical 5′ untranslated region (UTR) sequence or the complete 5′UTR sequence from c-JUN preceding the gene encoding the firefly luciferase (Fig. 5C). We compared the luciferase signals in HEK293T cells over-expressing eIF3D-WT or the R99K mutant, which mimic R99 methylation-deficient eIF3D (Fig. 5C) (Mutations in R97 of eIF3D does not yield soluble proteins, therefore we focused on R99 site for the functional studies). As shown in Fig. 5C, the firefly luciferase signals in the R99K mutant cell line were 23–34% lower in the translation of canonical mRNA and 47–62% lower for the mRNA containing c-JUN 5′UTR, compared with eIF3D-WT cells, suggesting that methylation-deficiency mutation in R99 affects the function of eIF3D as an important subunit of translation initiation complex in both the eIF4E-dependent canonical

translation pathway and the alternative translation pathway. The findings underscore the role of arginine methylation in eIF3D in cap-dependent translation initiation.

To further elucidate the role of eIF3D R99 methylation on the translation efficiency of endogenous genes utilizing these two translation initiation complexes in the cell, we performed polysome profiling using HEK293T cells overexpressing eIF3D-WT and the R99K mutant. RNA precipitation from each fraction followed by RT-qPCR showed a shift to lighter polysomes and monosomes in cells overexpressing R99K mutant for both ACTB and c-JUN, indicating a reduced association of their corresponding mRNA to ribosomes and thus lower translation efficiency (Fig. 5D). ACTB is known to utilize the canonical eIF4E-dependent translation initiation (Andreev et al, 2009).

Taken together, our results suggest that lack of methylation at R99 of eIF3D significantly affects the role of eIF3D as an important

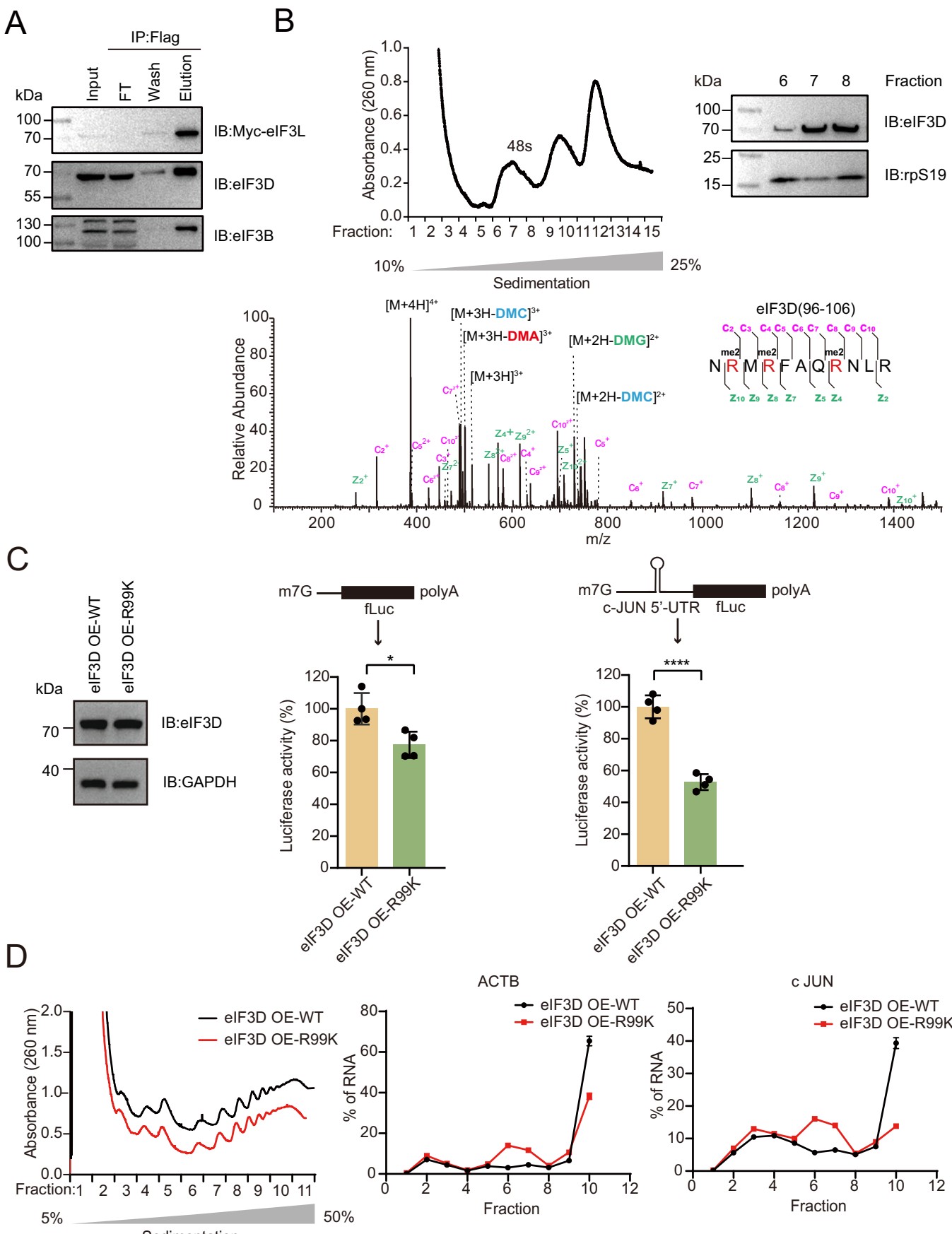

**Figure 5. Cap-dependent protein translation is regulated by the aDMA modification at R99/R103 of eIF3D.**

(A) Identification of dimethylated eIF3D in immunoprecipitated eIF3 complex. Plasmid constructs encoding the Full-length Flag-Myc-eIF3L proteins were transfected into HEK293T cells. eIF3L was immunoprecipitated with Flag agarose beads. The immunoprecipitated protein was analyzed by western blot. (B) Identification of dimethylated eIF3D in 48S ribosomal complexes. Fractionation of 48S ribosomal complexes by sucrose density gradient and western blot analysis of eIF3D and rpS19 (Top). MS/MS spectrum of isolated 48S identifies dimethylated eIF3D peptides (96–106) (Down). The spectrum was manually annotated with potential neutral losses. DMA, dimethylamine; DMC, dimethylcarbodiimide; DMG, dimethylguanidine. The abbreviations include their corresponding ions. (C) Luciferase activity in HEK293T cells stably expressing wild-type eIF3D and R99K mutant, using mRNAs carrying a canonical 5'UTR or the complete 5'UTR from c-JUN. Data are presented as mean ± SD; *$P = 0.0125$ and ****$P = 0.0000369$ by unpaired t test; $n = 4$. (D) Polysome association of c-JUN or ACTB mRNA in HEK293T cells stably expressing wild-type eIF3D and R99K mutant. mRNA abundance is calculated from RT-qPCR as a percentage of total transcript recovered from all fractions. Results are plotted as the mean ± SD; $n = 3$. Source data are available online for this figure.

subunit of the eukaryotic translation initiation factor in the eIF4E-dependent canonical translation initiation as well as the alternative translation initiation, resulting in reduced ribosomal binding to mRNAs and decreased translation efficiency. In other words, methylation at R99 of eIF3D plays a positive regulatory role in eukaryotic translation initiation. However, the molecular details regarding how exactly it is involved in the process require further structural analysis.

## Concluding remarks

Arginine methylation is a wide-spread PTM regulated by nine protein arginine methyltransferases (PRMTs) in human. The diverse and, in some cases, promiscuous substrate specificities of different PRMTs complicate the accurate prediction of functional methylarginine sites. RGG/RG-rich motifs, characterized by arginine residues flanked by one or more glycine residues, are well-established substrates for PRMTs (Bedford and Clarke, 2009). However, they are not the sole substrates. PRMT1, the primary enzyme for aDMA modification, methylates arginine residues in EGFR (Liao et al, 2015), EZH2 (Li et al, 2020), SMAD4 (Albrecht et al, 2018) and CDK4 (Dolezal et al, 2017), whose sites exhibit limited sequence conservation. PRMT4/CARM1 preferentially methylates arginine residues proximal to proline residues (Cheng et al, 2007; Shishkova et al, 2017). PRMT9 has two known substrates, SAP145/SF3B2 and MAVS, whose modification sites are located outside of both RGG/RG and RP motifs (Bai et al, 2022; Yang et al, 2015). This sequence diversity hinders the development of a pan-specific antibody for comprehensive enrichment of methylarginine-containing peptides. On the other hand, due to the subtle chemical differences between methylated and unmodified arginine residues, chromatography-based enrichment methods tend to identify peptides with multiple methylarginines, typified by RGG/RG motifs. Therefore, it is necessary to develop a sequence independent strategy, orthogonal to traditional antibody-based and chromatography-based enrichment methods.

We demonstrated in our study that the Tudor domain-based molecular affinity strategy could pan-specifically enrich methylarginines with mono-methylation and/or dimethylation in both RGG/RG motifs and non-RG motifs. Our strategy is complementary to existing methods and results in the identification of many new sites, thus enabling a more comprehensive understanding of cellular protein arginine methylation. It's worth noting that our affinity purification strategy requires a relatively small amount of sample input, compared to previous enrichment methods (Guo et al, 2014; Hartel et al, 2019; Ma et al, 2017; Uhlmann et al, 2012;

Wang et al, 2016), suggesting its potential utility in profiling physiologically relevant samples.

Using our molecular affinity strategy, we uncovered previous unknown function of arginine methylation in translation initiation factor eIF3D and demonstrated its regulation in protein translation. Beyond its established role in canonical cap-dependent protein translation, eIF3D has been implicated in alternative cap-dependent mRNA translation initiation, acting as a cap-binding protein for specific mRNAs, such as c-JUN (de la Parra et al, 2018; Lee et al, 2016). Our findings suggest that arginine methylation of eIF3D modulates this alternative cap-dependent protein translation pathway, potentially to a greater extent than the canonical pathway, as indicated by luciferase assays and polysome profiling (Fig. 5C,D). However, it is important to acknowledge that the functional studies directly compared wild-type eIF3D with the R99K mutant. While this comparison provides valuable insights into the functional role of arginine methylation, we must recognize the inherent limitations arising from the distinct biochemical properties of arginine and lysine residue. Consequently, further investigations are required to fully elucidate the regulatory mechanisms of arginine methylation in this process.

Furthermore, while our study, supported by both immunoblot and mass spectrometry analyses after immunoprecipitation, identified PRMT1 as a catalyst for eIF3D R99 dimethylation, it is important to consider the potential for functional redundancy among PRMTs. Other PRMTs within the cell could potentially contribute to the methylation of this same site, particularly in the absence of the primary 'housekeeping' enzyme, PRMT1. This also raises the possibility that the specific PRMT responsible for a given substrate's methylation may vary across different cellular contexts, thereby ensuring robust regulation of essential cellular functions.

## Methods

**Reagents and tools table**

| Reagent/Resource | Reference or Source | Identifier or Catalog Number |
|---|---|---|
| **Experimental models** | | |
| HEK293T cells (*H. sapiens*) | Cellcook Biotech | #CC4003 |
| HeLa cells (*H. sapiens*) | ATCC | #CCL-2 |
| **Recombinant DNA** | | |
| EZ-Tet-pLKO-Hygro | Addgene | #85972 |
| EZ-Tet-pLKO-Hygro-sheIF3D | This study | |
| EZ-Tet-pLKO-Puro | Addgene | #85996 |

| Reagent/Resource | Reference or Source | Identifier or Catalog Number |
|---|---|---|
| EZ-Tet-pLKO-Puro-shPRMT1 | This study | |
| pcDNA3.1 | Addgene | #138209 |
| pcDNA3.1-Cmyc-His6-eIF3D | This study | |
| pcDNA3.1-Cflag-His6-eIF3D | This study | |
| pcDNA3.1-Nflag-His6-PRMT1 | This study | |
| pcDNA3.1-Nflag-His6-PRMT2 | Kind gift from Wen Liu, Xiamen University, China | |
| pcDNA3.1-Nflag-His6-PRMT3 | Kind gift from Liu Wen, Xiamen University, China | |
| pcDNA3.1-Nflag-His6-PRMT4 | This study | |
| pcDNA3.1-Nflag-His6-PRMT6 | Kind gift from Wen Liu, Xiamen University, China | |
| pcDNA3.1-Nflag-His6-PRMT8 | This study | |
| pcDNA3.1-Nflag-PRMT5 | This study | |
| pcDNA3.1-NHis6-MEP50 | This study | |
| PET-28b (+) -CHis6-SMN Tudor WT (80–167aa) | This study | |
| PET-28b (+) -CHis6-SMN Tudor Y127L (80–167aa) | This study | |
| PET-28b (+) -CHis6-SMN Tudor N132E (80–167aa) | This study | |
| PET-28b (+) -CHis6-SMN Tudor W102F (80–167aa) | This study | |
| PET-28b (+) -NHalo-CHis6-SMN Tudor WT (80–167aa) | This study | |
| pGL4.13 [luc2/SV40] | Promega | #E6681 |
| pCMV-Cluc_2AG | New England Biolabs | #E2080S |
| pCMV-fLuc_2AG | This study | |
| pCMV-fLuc_c-JUN 5'UTR-2AG | This study | |
| PLVX-Puro | Takara | #632159 |
| PLVX-Puro-Cflag-eIF3D WT | This study | |
| PLVX-Puro-Cflag-eIF3D R99K | This study | |
| pMD2.G | Addgene | #12259 |
| psPAX2 | Addgene | #12260 |
| pSpCas9(BB)-2A-GFP | Addgene | #48138 |
| pSpCas9(BB)-2A-GFP-PRMT4KO | This study | |
| pSpCas9(BB)-2A-GFP-PRMT6KO | This study | |
| **Antibodies** | | |
| Mouse monoclonal anti-eIF3D | Santa Cruz Biotechnology | #sc-271515 |
| Mouse monoclonal anti-FLAG® M2 | Sigma | #F1804 |
| Mouse monoclonal anti-GAPDH | Santa Cruz Biotechnology | #sc-365062 |
| Mouse monoclonal anti-MYC | Cell Single Technology | #2276S |
| Mouse polyclonal anti-R99me2a | This study | |
| Rabbit monoclonal anti-PRMT1 | Abclonal | #A4502 |
| Rabbit monoclonal anti-PRMT4 | Abclonal | #A2246 |
| Rabbit monoclonal anti-PRMT6 | Abclonal | #A7814 |

| Reagent/Resource | Reference or Source | Identifier or Catalog Number |
|---|---|---|
| Rabbit monoclonal anti-Symmetric Dimethyl-arginine (SDMA) | Cell Single Technology | #13222 |
| Rabbit monoclonal anti-Asymmetric Dimethyl-arginine (ADMA) | Cell Single Technology | #13522 |
| Rabbit polyclonal anti-rpS9 | Abclonal | #A3675 |
| Rabbit polyclonal anti-eIF3B | Proteintech | #10319-1-AP |
| **Oligonucleotides and other sequence-based reagents** | | |
| PRMT4-targeting gRNA sequences 5'-CACCGATCATCA TCTCGGAGCCCAT-3' 5'-AAACATGGGCTC CGAGATGATGATC-3' | This study | |
| PCR primers for PRMT4 KO confirmation 5'-AGCTGACCGG CAGCAAAATTGT GAAGAGTAACA ACCTGACG-3' 5'-GAGCTGAAGC ACTGATGG-3' | This study | |
| PRMT6-targeting gRNA sequences 5'-CACCGGTGC TGCTGCGC TACAAAGT-3' 5'-AAACACTTTG TAGCGCAGC AGCACC-3' | This study | |
| PCR primers for PRMT6 KO confirmation 5'-AGCTGAC CGGCAGCAAAA TTGCTCCTCT ACCTGAACGAGC-3' 5'-CTCCCTCCC TAGAGGCTAT-3' | This study | |
| PRMT1-targeting shRNA sequences 5'-CCGGTTGAC TCCTACGCACA CTTTGCTCGAGC AAAGTGTGCGT AGGAGTCAA TTTTTG-3' 5'-AATTCAAA AATTGACTC CTACGCACA CTTTGCTCGA GCAAAGTG TGCGTAGG AGTCAA-3' | This study | |
| eIF3D-targeting shRNA sequences 5'-CTAGCGACG ACATGGATAA GAATGAACTCGA GTTCATTCT TATCCATGTCG TCTTTTTG-3' 5'-AATTCAAAAA GACGACATGG ATAAGAATGAACT CGAGTTCATTC TTATCCATGT CGTCG-3' | This study | |
| RT-qPCR primers for cJUN 5'-GTGCCG AAAAAGGA AGCTGG-3' 5'-GCTGCGTTA GCATGAGTTGG-3' | This study | |

| Reagent/Resource | Reference or Source | Identifier or Catalog Number |
| --- | --- | --- |
| RT-qPCR primers for ACTB 5'-GAGAAAAT CTGGCACC ACACC-3' 5'-GGATAGC ACAGCCTGG ATAGCAA-3' | This study | |
| **Chemicals, Enzymes and other reagents** | | |
| Protein A magnetic bead slurry | Genscript | #L00273 |
| RIPA Lysis Buffer | Genstar | #E121-01 |
| 0.25% Trypsin-EDTA (1X) | Gibco | #25200-072 |
| DMEM basic | Gibco | #C11995500BT |
| DMEM lacking L-methionine and L-cysteine | Gibco | #21013024 |
| Fetal bovine serum (FBS) | Gibco | #10270-106 |
| FreeStyle 293 Expression Medium | Gibco | #C12338018 |
| GlutaMAX (100X) | Gibco | #35050-061 |
| HEPES | Gibco | #15630130 |
| Phosphate-buffered saline (PBS) | Gibco | #10010023 |
| Penicillin/streptomycin | Gibco | #15140-122 |
| Bovine serum albumin (BSA) | Macklin | #B928042 |
| S-adenosylmethionine (AdoMet) | Macklin | #485-80-3 |
| Protease inhibitor cocktail (100X) | MCE | #HY-K0010 |
| L-Arginine | MedChemExpress | #HY-N0455 |
| Methylarginine (L-NMMA) | MedChemExpress | #HY-18732 |
| Asymmetric dimethylarginine (ADMA) | MedChemExpress | #HY-113216 |
| Symmetric dimethylarginine (SDMA) | MedChemExpress | #HY-101410 |
| Cycloheximide | MedChemExpress | #HY-12320 |
| Doxycycline, Hyclate | MedChemExpress | #24390-14-5 |
| MS023 | MedChemExpress | #HY-19615 |
| HaloLink™ Resin | Promega | #G1914 |
| Sequencing Grade Modified Trypsin | Promega | #V5111 |
| Hygromycin B | Sangon Biotech | #31282-04-9 |
| Puromycin | Sangon Biotech | #58-60-6 |
| Triton X-100 | Sangon Biotech | #9002-93-1 |
| IGEPAL® CA-630 | Sigma-Aldrich | #9002-93-1 |
| L-[12CD3] methionine | Sigma-Aldrich | #299154 |
| L-[12CH3] methionine | Sigma-Aldrich | #M5308 |
| Anti-DYKDDDDK Magnetic Agarose Beads | Thermo Fisher Scientific | #A36797 |
| PVDF Transfer Membranes, 0.45 µm | Thermo Fisher Scientific | #88518 |
| SuperSignal™ West Pico PLUS | Thermo Fisher Scientific | #34580 |
| Formic Acid, LC-MS Grade | Thermo Scientific Pierce | #85187 |
| Trifluoroacetic acid, LC-MS Grade | Thermo Scientific Pierce | #28903 |
| AGRGRGR, residues 96–102 of SmD1 | This study | |
| AGRme2sGRGR, residues 96–102 of SmD1 | This study | |
| AGRme2aGRGR, residues 96–102 of SmD1 | This study | |
| ARTQKTAYQRNRMRFAQRNLR, residues 86–106 of eIF3D | This study | |
| RapiGest SF Surfactant | Waters | #186001861 |
| Sep-Pak C18 1 cc Vac Cartridge | Waters | #WAT023590 |
| C18 membrane | 3 M | #2315 |
| C8 membrane | 3 M | #2214 |
| Mouse IgG | Beyotime | #A7028 |
| Polybrene | Beyotime | #C0351 |
| RVC | Beyotime | #R0108 |
| RNase inhibitor | Beyotime | #R0102 |
| BCA assay | DINGGUO | #BCA02 |
| Firefly Luciferase Reporter Gene Assay Kit | Beyotime | #RG005 |
| HiScript III Reverse Transcriptase | Vazyme | #R323-01 |
| Phusion™ High-Fidelity DNA Polymerase | Thermo Fisher Scientific | #F530S |
| PierceTM Quantitative Colorimetric Peptide Assay kit | Thermo Fisher Scientific | #23275 |
| Polyethylenimine, Linear, MW 25000, Transfection Grade (PEI 25 K™) | Polysciences | #26913-06-4 |
| PrimeSTAR HS DNA Polymerase | Takara | #R010A |
| RNAiso Plus | Takara | #9108 |
| Taq Pro Universal SYBR qPCR Master Mix | Vazyme | #Q712-02 |
| T7 mRNA Kit | New England Biolabs | #E2080S |
| Lipofectamine™ 3000 | Thermo Fisher Scientific | #L3000001 |
| Lipofectamine™ MessengerMAX™ | Thermo Fisher Scientific | #LMRNA001 |
| HB-Infusion™ | HanBio | #HB-infusion-20T |
| **Software** | | |
| Flexanalysis | BRUKER | https://massspec.chem.ox.ac.uk/flexanalysis |
| GraphPad Prism v9.5.0 | GraphPad | https://www.graphpad.com/ |
| MicroCal PEAQ-ITC Analysis Software | MICROCAL | https://www.malvernpanalytical.com/en/support/product-support/software/microcal-peaq-itc-analysis-software-v141 |
| MicroCal PEAQ-ITC Control Software | MICROCAL | https://www.malvernpanalytical.com.cn/support/product-support/software/microcal-peaq-itc-control-software-update-v141 |
| Proteome Discoverer 2.2 | Thermo Fisher Scientific | https://www.thermofisher.cn/order/catalog/product/OPTON-31014?SID=srch-srp-OPTON-31014 |

| Reagent/Resource | Reference or Source | Identifier or Catalog Number |
|---|---|---|
| PyMOL 3.0 | Schrodinger | https://www.schrodinger.com/products/pymol |
| Xcalibur | Thermo Fisher Scientific | https://www.thermofisher.cn/order/catalog/product/OPTON-30965?SID=srch-srp-OPTON-30965 |
| **Critical commercial assays** | | |
| BCA assay | DINGGUO | #BCA02 |
| Firefly Luciferase Reporter Gene Assay Kit | Beyotime | #RG005 |
| HiScript III Reverse Transcriptase | Vazyme | #R323-01 |
| Phusion™ High-Fidelity DNA Polymerase | Thermo Fisher Scientific | #F530S |
| PierceTM Quantitative Colorimetric Peptide Assay kit | Thermo Fisher Scientific | #23275 |
| Polyethylenimine, Linear, MW 25000, Transfection Grade (PEI 25 K™) | Polysciences | #26913-06-4 |
| PrimeSTAR HS DNA Polymerase | Takara | #R010A |
| RNAiso Plus | Takara | #9108 |
| Taq Pro Universal SYBR qPCR Master Mix | Vazyme | #Q712-02 |
| T7 mRNA Kit | New England Biolabs | #E2080S |
| Lipofectamine™ 3000 | Thermo Fisher Scientific | #L3000001 |
| Lipofectamine™ MessengerMAX™ | Thermo Fisher Scientific | #LMRNA001 |
| HB-Infusion™ | HanBio | #HB-infusion-20T |
| **Other** | | |
| Microcal PEAQ-ITC calorimeter | Malvern | |
| Orbitrap Fusion Lumos Tribrid mass spectrometer | Thermo Fisher Scientific | |
| MALDI time-of-flight (TOF) mass spectrometer | Bruker | |
| GloMax® Navigator microplate luminometer | Promega | |

## Cell culture and reagents

HEK293T (Cellcook Biotech) and HeLa cells (ATCC) were maintained in DMEM (Gibco) supplemented with 10% fetal bovine serum (FBS, Gibco) and 1x Pen-Strep (Gibco) at 37 °C under 5% $CO_2$ atmosphere. These cell lines were authenticated by STR profiling and tested for mycoplasma contamination. For plasmid transfection, cells grown to about 80% confluence were transfected with Polyethylenimine (Polysciences) according to the manufacturer's instructions. MS023 were obtained from MedChemExpress and used at 1 μM for 3 days unless otherwise specified.

## Plasmid construction

The DNA encoding SMN Tudor domain (80–167 aa) was synthesized by Qingke and cloned into pReceiver-B80.1a via BstBI and NotI restriction sites, which expresses SMN Tudor with an N-terminal Halo tag and C-terminal His$_6$-tag. For ITC experiment, the DNA encoding SMN Tudor domain was subcloned into PET-28b (+) plasmid via NdeI and XhoI restriction sites, which expresses SMN Tudor with a C-terminal His$_6$-tag. To generate expression constructs for SMN Tudor Y127L, W102F and N132E mutant, site-directed mutagenesis was performed using the Phusion™ High-Fidelity DNA Polymerase (Thermo Fisher Scientific), according to the manufacturer's instruction.

The full-length DNA encoding human PRMT5 (NM_006109.5) with an N-terminal Flag-tag was synthesized by Qingke, and inserted into the pcDNA3.1 vector via NheI and BamHI restriction sites. The full-length DNA encoding human MEP50 (NM_006109.5) was synthesized by Qingke, and inserted into the pcDNA3.1 vector with an N-terminal His$_6$-tag via NotI and XbaI restriction sites. The full-length DNA encoding human PRMT2 (NM_206962.4), PRMT3 (NM_005788.4) and PRMT6 (NM_018137.3) were generous gifts from Professor Liu Wen, and were cloned into pcDNA3.1 vector with an N-terminal Flag-tag and His$_6$-tag via KpnI and XhoI restriction sites. The full-length DNA encoding human PRMT1 (NM_001536.6), PRMT4 (NM_199141.2) and PRMT8 (NM_019854.5) was synthesized by Qingke, and inserted into the pcDNA3.1 vector with an N-terminal Flag-tag and His$_6$-tag via NotI and HindIII restriction sites. All of the above plasmids were used to purify recombinant enzymes and for mammalian expression.

The full-length DNA encoding human eIF3D was PCR-amplified from HEK293T cDNA and cloned into pcDNA3.1 vector with a C-terminal Flag-tag and His$_6$-tag or a C-terminal Myc-tag and His$_6$-tag via AFIII and XhoI restriction sites, respectively. To generate the expression constructs for eIF3D methylation site mutant R99K, site-directed mutagenesis was performed using the Phusion™ High-Fidelity DNA Polymerase (Thermo Fisher Scientific), according to the manufacturer's instruction.

The full-length DNA encoding human eIF3L was PCR-amplified from HEK293T cDNA and cloned into pcDNA3.1 vector with a C-terminal Flag-tag and Myc-tag via HindIII and NheI restriction sites, respectively.

To generate the luciferase reporter plasmid, the gene encoding firefly luciferase was amplified from the pGL4.13 [luc2/SV40] vector (Promega) and inserted into pCMV-Cluc_2AG vector (New England Biolabs) via NotI and BamHI restriction sites. The resulting plasmid was named pCMV-fLuc_2AG. To generate the luciferase reporter plasmid with the complete 5′UTR from c-JUN, the c-JUN 5′UTR was amplified from HEK239T cDNA, and inserted into the pCMV-fLuc_2AG via HindIII and BamHI restriction sites. All the constructs were confirmed by DNA sequencing.

## hmSILAC cell culture

HeLa cells were cultured in DMEM lacking L-methionine and L-cysteine (Thermo Fisher Scientific). Light and heavy hmSILAC media were supplemented with L-[$^{12}CH_3$] methionine (Met-0, light, Sigma-Aldrich) and L-[$^{13}CD_3$] methionine (Met-4, heavy, Sigma-Aldrich) at 30 mg/L, respectively. The hmSILAC media were further supplemented with 10% FBS, 1 x Pen-Strep, 1% GlutaMax (Gibco), 1 mM sodium pyruvate (Gibco), L-Cysteine dihydrochloride (63 mg/L, sigma), and 10 mM HEPES (pH 7.5, Gibco). HeLa cells were cultured in hmSILAC media for 15 replication cycles

## SMN Tudor domain expression and purification

The expression plasmids for SMN Tudor domain and Halo-Tudor fusion protein were separately transformed into BL21(DE3) competent cells and cultured in LB broth medium with 50 µg/mL Kanamycin at 37 °C until the $OD_{600}$ reached 0.6–0.8. Protein expression was induced by 0.5 mM IPTG at 37 °C for 4 h and cells were harvested. Cell pellets were resuspended in lysis buffer containing 50 mM Tris-HCl, pH 8, 500 mM NaCl, and 10 mM imidazole and sonicated on ice. The supernatant was collected after centrifugation at 21,000 rpm for 30 min and protein purification was performed by Ni-NTA chromatography (TransGen Biotech). The proteins were eluted using elution buffer (50 mM Tris-HCl, pH 8, 500 mM NaCl, 40 mM imidazole) and the eluents were analyzed by SDS-PAGE. Purified Halo-Tudor fusion protein was dialyzed overnight in 100 mM Tris-HCl, pH 7.6, and 150 mM NaCl for subsequent affinity resin preparation. SMN Tudor protein used in the ITC assay was dialyzed overnight in PBS and purified again using Superdex-75 size exclusion column.

## ITC experiment

Both protein (SMN Tudor and mutants) and ligands (MMA, SDMA, ADMA, Arginine and SmD1-derived heptapeptides) were dissolved in 1×PBS, and the Microcalorimetric experiment was determined by Microcal PEAQ-ITC calorimeter (Malvern). The peptides used were SmD1 (AGRGRGR), SmD1-sDMA ($AGR_{me2s}GRGR$), SmD1 aDMA ($AGR_{me2a}GRGR$). A 200 µL sample cell was filled with 200 µM or 300 µM protein solution, and a 70 µL micro-syringe was filled with 2 mM or 5 mM ligand solution. The reference cell was filled with $H_2O$. Using MicroCal PEAQ-ITC Control Software to control each 2 µL injection of ligand containing the sample in microsyringe with an interval of 150 s at 25 °C with 500 rpm stirring speed for 20 times in total. The ITC data were processed by Microcal Peaq-ITC Analysis software with one site model.

## Preparation of Tudor affinity resin

100 µL suspension of 25% HaloLink agarose resin slurry (Promega) in 25% ethanol was transferred to a 1.5 mL low-binding tube. The resin was washed three times with 500 µL ultrapure water to remove ethanol followed by equilibration with 3 × 500 µL solution (100 mM Tris-HCl, pH 7.6, 150 mM NaCl, 0.05% IGEPAL ®CA-630). 400 µg Halo-Tudor fusion protein was added to the resin and incubated at room temperature for 2.5 h. After centrifugation, the supernatant was discarded and the resin was washed with 3 × 500 µL wash buffer (100 mM Tris-HCl, pH 7.6, 150 mM NaCl, 1 mg/ml BSA, 0.05% IGEPAL ®CA-630). Finally, the Tudor affinity resin was resuspended to the required volume of buffer (50 mM HEPES, pH 7.2, 10 mM $Na_2HPO_4$, 50 mM NaCl) for the downstream applications.

## Methylarginine enrichment by Tudor affinity resin

HeLa cells were lysed in 50 mM ammonium bicarbonate, 0.2% RapiGest SF Surfactant (Waters), and the lysate was homogenized by sonication. Cleared lysates were diluted in 50 mM ammonium bicarbonate to bring the final concentration of RapiGest below 0.2% before being subjected to reduction with DTT, alkylation with iodoacetamide, and digestion with trypsin (Thermo Fisher Scientific). The tryptic digest of HeLa lysate was purified using Sep-Pak C18 columns (Waters) and redissolved in 200 µL binding buffer (50 mM HEPES, pH 7.2, 10 mM $Na_2HPO_4$, 50 mM NaCl). The peptide concentration was determined by Pierce™ Quantitative Colorimetric Peptide Assay kit (Thermo Fisher Scientific). 360 µg peptide and 100 µL Tudor affinity resin were co-incubated at 4 °C for 2.5 h. Then, the resin was washed twice with 500 µL binding buffer followed by one wash with 500 µL precooled water and eluted with 3 × 70 µL 0.15% TFA. Peptide eluates were desalted on reverse-phase C18 StageTips (Sigma) before being subjected to LC-MS analysis.

## Mass spectrometry analysis

Samples were analyzed on an EASY-nLC 1200 system coupled to an Orbitrap Fusion Lumos Tribrid mass spectrometer equipped with a Nanospray Flex™ Ion Source (Thermo Fisher Scientific, San Jose, CA). Mobile phase A: water with 0.1% FA. Mobile phase B: 80% acetonitrile, 0.1% FA, and 19.9% water. Nano-LC was operated in a single analytical column, packed in-house with Reprosil-Pure-AQ C18 phase (Dr. Maisch, 1.9 µm particle size, 15 cm column length), at a flow rate of 200 nL/min. All samples dissolved in 0.1% FA were injected onto the column and eluted in a 2 h gradient. The nanoSpray ion source was operated at 2.2 kV spray voltage and 275 °C heated capillary temperature. The mass spectrometer was set to acquire full scan MS spectra (300–1750 $m/z$) for a maximum injection time of 120 ms at a mass resolution of 60,000 and an automated gain control (AGC) target value of 1e6. The dynamic exclusion was set to 40 s at an exclusion window of 10 ppm. In HCD or ETciD scans, the collision energy was set at 27% or 30% respectively, utilizing the fixed collision energy mode. In all HCD/ETciD scans, AGC target was set to $2.0e^5$ and maximum injection time was 86 ms. For ETD scans, calibrated charge dependent ETD parameters was set as True. All MS/MS spectra were acquired in the Orbitrap with resolution at 50,000 in profile mode.

## Data analysis

MS data processing for all raw files was performed using Proteome Discoverer (PD) version 2.2 software (Thermo Fisher Scientific) and further data analysis was done in R with in-house scripts. Raw files were searched with Sequest HT search engine against a human-specific database (Uniprot reference proteome retrieved from January 2023, containing 20,422 canonical entries). Enzyme restriction was set to trypsin digestion with full specificity (C-terminal to lysine and arginine) and a maximum of 5 missed cleavages. The precursor mass tolerance was set to 10 ppm and fragment ion mass tolerance to 0.02 Da. Carbamidomethylation on cysteine residues was used as a fixed modification. Methionine oxidation, mono-methyl-R and di-methyl-R were used as variable modifications during raw data processing, with the maximum number of variable modifications set to 5. FDR threshold was set at 1% at both peptide and protein levels in Peptide and Protein FDR Validator node. Sites with localization probabilities >=98% using ptmRS Node in PD 2.4 were defined as unambiguous sites. Putative

arginine methylated proteins were used for Gene ontology (GO) analysis using DAVID Bioinformatics Resources version 2021. For hmSILAC data, in the "light" analysis, monomethylation of R (+14.016 Da), dimethylation of R (+28.031 Da) and oxidation of M (+15.995 Da) were specified as variable modifications; carbamidomethylation of Cys (+57.021 Da) was specified as a fixed modification. In the "heavy" analysis, heavy monomethylation of R (Methyl4, +18.038 Da), heavy dimethylation of R (Dimethyl4, +36.076 Da), heavy methionine (Met4, +4.022 Da) and oxidized heavy methionine (OxMet4, +20.017 Da) were specified as variable modifications; carbamydomethylation of Cys (+57.021 Da) was specified as a fixed modification. Identification of light and heavy methyl-peptide doublets were processed together and integrated using a customized pipeline developed in R. This pipeline identified doublets of heavy and light hmSILAC peptides from output tables of peptide groups in PD. Only peptide precursors identified in either light or heavy forms were subjected to the analysis. If heavy and light methyl peptide pairs were considered a true doublet, a pair of peaks must satisfy all the following conditions: (1) m/zH > m/zL. (2) ChargeH = ChargeL. (3) |RTH – RTL| < Delta RT threshold (default = 0.5 min). (4) |ME| < ME| (mass error) threshold (default = 2 ppm). (5) |Log$_2$(H/L ratio)| < Log$_2$Ratio threshold (default = 1.6) (Massignani et al, 2019).

## Expression and purification of PRMT1, 2, 3, 4, 6, 8 and PRMT5/MEP50

HEK293T cells were maintained in DMEM Medium with 10% FBS and transfected with PRMT1, PRMT2, PRMT3, PRMT4, PRMT6, PRMT8 vector or PRMT5/MEP50 vectors using Lipofectamine 3000 (Thermo Fisher Scientific) according to the manufacturer's instructions. Cells were collected 48–72 h post transfection, washed and resuspended in the binding buffer (50 mM Tris-HCl, pH 7.5, 300 mM NaCl, 10 glycerol, 20 mM imidazole, 1 mM tris (2-carboxyethyl) phosphine-HCl (TCEP), 0.1% Triton X-100 and Protease Inhibitor Cocktail). The cells were lysed by sonication for 2 min and cleared by centrifugation at 15,000 × g for 10 min at 4 °C. The supernatant was subjected to protein purification through IMAC affinity chromatography (GE). The proteins were eluted using elution buffer (50 mM Tris-HCl, pH 7.5, 300 mM NaCl, 10% glycerol, 200 mM imidazole, 1 mM TCEP, 0.1% Triton X-100) and the eluents were analyzed by SDS-PAGE. Purified proteins were pooled, concentrated and stored at −80 °C.

## In vitro methylation reactions

A 20 μL reaction containing 2.5 μM peptide (ARTQKTAYQRNRMR-FAQRNLR, residues 86–106 of eIF3D) and 100 μM S-adenosylmethionine (AdoMet) (Macklin) in the methylation buffer (50 mM Tris-HCl, pH 8.5, 1 mM DTT) were initiated by adding purified PRMT5-MEP50 or PRMT1 to a final concentration of 150 nM and incubated at 25 °C for 12 h. And 300 nM purified PRMT2/3/4/6/8 protein was incubated with 2.5 μM peptide in methylation buffer (50 mM Tris-HCl, pH 8.5, 0.5 mM EDTA, 100 mM NaCl) with the presence of 100 μM S-adenosylmethionine at 30 °C for 12 h in a final volume 20 μL. Reactions were terminated by the addition of 3 μL FA. Peptide were desalted on reversed-phase C18 StageTips before analysis on a MALDI time-of-flight (TOF) mass spectrometer (Bruker, Leipzig, Germany).

## Development of anti-R99me2a antibody

The rat polyclonal antibody anti-R99me2a was developed against the modified peptide RNRMR$_{me2a}$FAQRNL by Atagenix Technology Co., Ltd. The process began with the design of both aDMA modified and unmodified immunogenic peptides (Cys-RNRMR$_{me2a}$FAQRNL, and Cys-RNRMRFAQRNL). Two rats were immunized with the modified peptide conjugated to keyhole limpet hemocyanin (KLH) over a period of approximately 45 days, with a total of five immunization rounds. The resulting immune serum was evaluated for titer using an indirect enzyme-linked immunosorbent assay (ELISA). The immune serum with a titer surpassing 1:32,000 was harvested and underwent affinity purification. The specificity of the purified antibodies was validated through dot blot experiments against the aDMA modified and unmodifed peptides. The final yield of the usable antibody was 0.16 mg, with a concentration of 0.81 mg/mL. The antibody was aliquoted and stored at −20 °C for future use.

## Generation of eIF3D-overexpressing stable cell lines

To generate stable cell lines with inducible knockdown of eIF3D, we used the EZ-Tet-pLKO-Hygro vector (Addgene) by inserting the shRNA sequence listed in Reagents and tools table. For lentiviral vectors packaging and infection, HEK293T cells were seeded in 60 mm culture plates and transfected with lentiviral vectors together with packaging vectors (5 μg EZ-Tet-pLKO-Hygro vector, 1.25 μg pMD2.G (Addgene) and 3.75 μg psPAX2 (Addgene)) using 30 μL PEI for 48 h according to the manufacturer's protocol. Virus production was allowed to proceed for 48 h. Lentiviral media was then collected from each plate, and centrifuged at 2000 rpm for 5 min to clear any cellular debris. Lentiviral media was then subpackaged to a 15 mL centrifuge tube and stored at −80 °C until the day of infection. HEK293T cells were infected with the lentiviruses produced for 48 h and seeded into 96-well plates at a density of 0.5 cells per well. Selection with 75 μg/ml hygromycin B (31282-04-9, Sangon Biotech) was performed for 10–14 days, with the medium containing fresh antibiotics replaced every 3 days. To induce knockdown of eIF3D in the stable cell lines, cells were treated with doxycycline (MedChemExpress) at 1.25 μg/mL for 72 h. Single colonies were subjected to immunoblotting analysis to confirm knockdown.

The eIF3D sequence for overexpression the WT and R99K mutant was generated by making synonymous mutations (gacga-catggataagaatgaa→ gaTgaTatggaCaaAaaCgaG) to the shRNA binding site and was inserted into the PLVX-Puro vector (Takara) with a C-terminal Flag-tag. The lentiviral vectors packaging process is as described above, with the lentiviral vector being PLVX-Puro vector. HEK293T cells with inducible knockdown of eIF3D were infected with the lentiviruses produced and selected for stable cells with 2 μg/mL Puromycin (Sangon Biotech) for 2 days.

## Generation of HEK293T cell lines with PRMT1 knockdown

To generate cell lines with knockdown of PRMT1, we used the EZ-Tet-pLKO-Puro vector (Addgene) by inserting the shRNA sequence listed in Reagents and tools table. The lentiviral vectors packaging process is as described above, with the lentiviral vector

being EZ-Tet-pLKO-Puro vector. HEK293T cells were infected with the lentiviruses produced and selected for stable cells with 2 µg/mL Puromycin for 2 days.

## Generation of PRMT4 and/or PRMT6 KO HEK293T cell lines

gRNAs targeting exon 6 of PRMT4 and exon 1 of PRMT6 were designed and cloned into vector pSpCas9(BB)-2A-GFP (Addgene), respectively. gRNA plasmids were transfected into HEK293T cells using Lipofectamine 3000 and high GFP expression cells were enriched by fluorescence-activated cell sorting 48 h after transfection, followed by 7 days of further culturing before single cell sorting into 96-well plates. Cell clones with frameshift mutations were identified by Indel Detection by Amplicon Analysis (IDAA) (Steentoft et al, 2013) using the primers in Reagents and tools table. Gene knockouts were confirmed by Sanger sequencing. PRMT4 and PRMT6 double knockout HEK293T cells were generated by further knocking out PRMT6 in PRMT4 knockout HEK293T cells, following the same protocol.

## Immunoprecipitation and western blot analysis

Cells were lysed in lysis buffer (50 mM Tris pH 7.4, 150 mM NaCl, 0.5% CA-630, 1 mM $Na_3VO_4$ and protease inhibitor cocktail). Cell lysates were incubated with anti-Myc antibody (2276, Cell Signaling Technology) or anti-Flag antibody (F1804, sigma) at 4 °C overnight. Subsequently, 20 µL of 80% Protein A magnetic bead slurry (L00273, Genscript) was added, and the incubation was continued at 4 °C for 3.5 h. The supernatant was removed after centrifugation and the resin was washed with PBS. The immunocomplex was eluted with elution buffer (2% SDS and 10 mM DTT), resolved by SDS-PAGE and transferred to PVDF membranes (88518, Thermo Fisher Scientific). Membranes were blocked for 1 h in 5% non-fat milk (BioFoxx) at room temperature, and then incubated with primary antibody at 4 °C overnight and secondary antibodies for 2 h at room temperature, respectively. Primary antibodies used for western blot analysis were: anti-aDMA (13522, Cell Signaling Technology, 1:1000), anti-sDMA (13222, Cell Signaling Technology, 1:1000), anti-eIF3D (sc-271515, Santa Cruz Biotechnology, 1:1000), anti-R99-me2a (atagenix, 1:300), anti-GAPDH (sc-365062, Santa Cruz Biotechnology, 1:3000), anti-PRMT1 (A4502, Abclonal, 1:1000), anti-PRMT4 (A2246, Abclonal, 1:1000), anti-PRMT6 (A7814, Abclonal, 1:1000), anti-Flag (F1804, Sigma, 1:1000), anti-Myc (2276, Cell Signaling Technology, 1:3000), anti-rpS19 (A3675, Abclonal, 1:1000), and anti-eIF3B (10319-1-AP, Proteintech, 1:500).

## Immunoprecipitation and mass spectrometry

To confirm aDMA modifications at R99 and R103 of eIF3D, Flag-tag recombinant eIF3D protein purified from HEK293T cells was resolved on 10% SDS–PAGE and stained with Coomassie blue R-250. The eIF3D band was excised, washed with Milli-Q water and de-stained with 100 mM ammonium bicarbonate/acetonitrile (1:1, vol/vol) for 3 h, followed by dehydration in acetonitrile, rehydration with 10 mM DTT for 45 min at 56 °C and with 55 mM iodoacetamide for 20 min at room temperature in the dark. Then added enough buffer (10 mM ammonium bicarbonate containing

10% (vol/vol) acetonitrile) to cover the dry gel pieces, placed gel pieces in ice for 2 h to saturate them with trypsin. Proteins were digested with trypsin at 37 °C for 16 h.

To identify arginine methylation sites of eIF3D in the translation initiation complex, we transfected HEK293T cells in two 10 cm dishes with 12 µg pcDNA3.1-Cter-Flag-Myc-eIF3L-hs plasmid for 48 h, the cell pellets were lysed in lysis buffer (50 mM HEPES-KOH pH 7.5, 150 mM KCl, 5 mM $MgCl_2$, 0.5% NP-40, protease inhibitor cocktail) for 30 min on ice. Centrifuge at $15,000 \times g$ for 10 min at 4 °C, and then transfer the supernatant to a new tube. The lysate was incubated with 40 µL Pierce™ Anti-DYKDDDDK Magnetic Agarose Beads (A36797, Thermo Fisher Scientific) overnight at 4 °C. Beads were washed three times with high salt wash buffer (50 mM HEPES-KOH pH 7.5, 500 mM KCl, 5 mM $MgCl_2$, 0.5% NP-40), and then eluted with 100 µL elution buffer (100 mM Glycine pH 2.8). eIF3 complex was subjected to SDS-PAGE followed by Western blot and mass spectrometry analysis. Elution was neutralized with 1 M Tris-HCl, pH 9, follow by reduction with 10 mM DTT, alkylation with 40 mM iodoacetamide, and digestion with 1 µg trypsin at 37 °C for 4 h.

To validate the functional role of PRMT1 in eIF3D R99 methylation, immunoprecipitation of eIF3D from HEK293T cells was performed with or without PRMT1 knockdown or MS023 treatment, followed by the procedures described above.

For LC-MS/MS analysis. the resulting peptides from immunoprecipitated eIF3D were desalted using reverse-phase C18 Stage-Tips according to the manufacturer's instructions. Next, the samples were evaporated in a vacuum centrifuge and then resuspended in 0.1% FA and subjected to liquid chromatography coupled with tandem mass spectrometry (LC-MS/MS). The samples were detected using both HCD only and ETciD method.

## Isolation of 48S ribosomal complexes

48S ribosomal complexes were isolated as previously described (Lee et al, 2016; Wilson et al, 2000). Briefly, three 100 mm dishes of fresh cells were treated with 0.1 mg/mL Cycloheximide (CHX, MedChemExpress) at 37 °C and 5% $CO_2$ for 5 min. Then, the cells were washed twice with ice-cold PBS containing 0.1 mg/ml CHX. The cell pellet was resuspended in lysis buffer (20 mM HEPES pH 7.6, 150 mM KOAc, 2.5 mM Mg(OAc)$_2$, 5 mM DTT, 0.5% Triton X-100, 0.1% protease inhibitor cocktail, 0.5 U/µL RNase inhibitor, 0.1 mg/ml CHX, 1 mM GMP-PNP and lysed on ice for 20 min. Cells were disrupted by grinding 10–20 times with a 1 mL syringe. Lysate was centrifuged at 13,000 rpm for 10 min at 4 °C, and the supernatant was transferred to a new tube. 100 ng of RNA was added to the supernatant and incubated at 30 °C for 20 min. The reaction mixture was fractionated through a 10–25% (w/v) sucrose gradient by centrifugation for 3.5 h at 38,000 rpm at 4 °C with Beckman SW41 Ti rotor. Fractions were collected from the top of the sucrose gradient using Biocomp, with 15 tubes for collection. Protein was precipitated with trichloroacetic acid and resuspended in 50 mM ammonium bicarbonate, 0.1% Rapigest. Lysate was subjected to reduction with 10 mM DTT, alkylation with 40 mM iodoacetamide, and digestion with 1 µg trypsin at 37 °C for 5 h. The resulting peptides were desalted using reverse-phase C18 StageTips according to the manufacturer's instructions. Next, the samples were evaporated in a vacuum centrifuge and resuspended in 0.1% FA and subjected to liquid

chromatography coupled with tandem mass spectrometry using ETciD method.

## Luciferase reporter assay

The reporter plasmids were linearized by the XbaI and mRNAs were produced by in vitro transcription with T7 mRNA Kit (New England Biolabs) according to the manufacturer's instructions, which included the addition of a 5′ cap.100 ng mRNAs were transfected into HEK293T cells Lipofectamine™ MessengerMAX™ (Thermo Fisher Scientific) according to the manufacturer's instructions. Cells were harvested 18 h after transfection and firefly luciferase activities were measured using Firefly Luciferase Reporter Gene Assay Kit (Beyotime) on a GloMax® Navigator microplate luminometer (Promega).

## Polysome profiling

Polysome Profiling was performed as previously described (Choudhuri et al, 2013; Han et al, 2022). Briefly, two 100 mm dishes of fresh cells were treated with 0.1 mg/mL CHX at 37 °C and 5% $CO_2$ for 5 min. Then, cells were washed twice with ice-cold PBS containing 0.1 mg/ml CHX. The cell pellet was resuspended in lysis buffer (10 mM Tris-HCl, pH 8, 140 mM NaCl, 5 mM $MgCl_2$, 1% Triton X-100, 0.5% sodium deoxycholate, 0.4 U/μL RNase inhibitor, 10 mM RVC, 0.1% protease inhibitor cocktail, 20 mM DTT, 0.1 mg/ml CHX) and kept on ice for 20 min. Lysate was centrifuged at 13,000 rpm for 10 min at 4 °C, and the supernatant was transferred to a new tube. The supernatant was further centrifuged through a 5–50% (w/v) sucrose gradient at 36,000 rpm, 4 °C for 2 h with a Beckman SW41 Ti rotor. Fractions were collected from the top of the sucrose gradient using Biocomp. From the appropriate fractions, RNA was purified by phenol-chloroform extraction and ethanol precipitation.

## Reverse transcription and quantitative PCR (RT-qPCR)

Purified RNA was reverse transcribed to cDNA using HiScript III Reverse Transcriptase (Vazyme) according to the manufacturer's instruction, and qPCR was performed using Taq Pro Universal SYBR qPCR Master Mix (Vazyme) in a CFX ConnectTM RT-PCR Detection System (BioRad Laboratories). Housekeeping gene GAPDH was used as an internal control for normalization. The primer sequences for RT-qPCR were listed in Reagents and tools table.

# Data availability

The mass spectrometry proteomics data have been deposited to the ProteomeXchange Consortium via the PRIDE partner (Vizcaino et al, 2016) repository with the dataset identifier PXD053390 (https://proteomecentral.proteomexchange.org/cgi/GetDataset?ID=PXD053390).

The source data of this paper are collected in the following database record: biostudies:S-SCDT-10_1038-S44319-025-00599-y.

# Peer review information

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

## Acknowledgements

We thank Prof. Wen Liu for providing plasmids encoding human PRMT2, PRMT3 and PRMT6. This work was supported by grants from the National Natural Science Foundation of China (32271497, 92478126, and T2450074), the National Key Research and Development Program, Ministry of Science and Technology of China (2021YFA1200903), Guangdong Provincial Key Laboratory of Drug Non-Clinical Evaluation and Research (2023B1212070029) and the China Postdoctoral Science Foundation (2025M773561). Zilu Ye was supported by the Non-profit Central Research Institute Fund of the Chinese Academy of Medical Sciences (grant no. 2023-RC180-03), the Chinese Academy of Medical Sciences (CAMS) Innovation Fund for Medical Sciences (grant no. 2022-I2M-2-004, 2023-I2M-2-005, 2021-I2M-1-061), National

Natural Science Foundation of China (22574120), Natural Science Foundation of Jiangsu Province (BK20240443), the Suzhou Municipal Key Laboratory (SZS2022005) and the NCTIB Fund for the R&D Platform for Cell and Gene Therapy.

## Author contributions

**Lingzi Lu**: Conceptualization; Data curation; Software; Formal analysis; Validation; Investigation; Visualization; Methodology; Writing—original draft; Writing—review and editing. **Ting Li**: Data curation; Formal analysis; Validation; Investigation; Visualization; Methodology; Writing—original draft. **Rou Zhang**: Conceptualization; Data curation; Formal analysis; Validation; Investigation; Visualization; Methodology; Writing—original draft. **Yutong Wang**: Data curation; Formal analysis; Validation; Visualization; Methodology; Writing—original draft. **Xiaoping Ye**: Data curation; Formal analysis; Investigation; Methodology. **Yixin Luo**: Data curation; Formal analysis. **Lingyu Sun**: Data curation; Formal analysis; Funding acquisition. **Liang Qi**: Resources. **Zilu Ye**: Conceptualization; Resources; Data curation; Software; Formal analysis; Supervision; Validation; Visualization; Methodology; Writing—original draft. **Yang Mao**: Conceptualization; Resources; Supervision; Funding acquisition; Investigation; Methodology; Writing—original draft; Project administration; Writing—review and editing. **Yanqiu Yuan**: Conceptualization; Resources; Data curation; Software; Formal analysis; Supervision; Funding acquisition; Validation; Investigation; Visualization; Methodology; Writing—original draft; Project administration; Writing—review and editing.

Source data underlying figure panels in this paper may have individual authorship assigned. Where available, figure panel/source data authorship is listed in the following database record: biostudies:S-SCDT-10_1038-S44319-025-00599-y.

## Disclosure and competing interests statement

The authors declare no competing interests.

# Expanded View Figures

**Figure EV1.   ITC titrations of SMN Tudor protein with SmD1-derived heptapeptides or methylarginines.**

The lines represent data fitting curves using 1:1 binding model at a fixed stoichiometry (N) of 1. The peptides used were SmD1 (AGRGRGR), SmD1-sDMA (AGR$_{me2s}$GRGR), SmD1 aDMA (AGR$_{me2a}$GRGR).

▶

                                                                                                                                                      

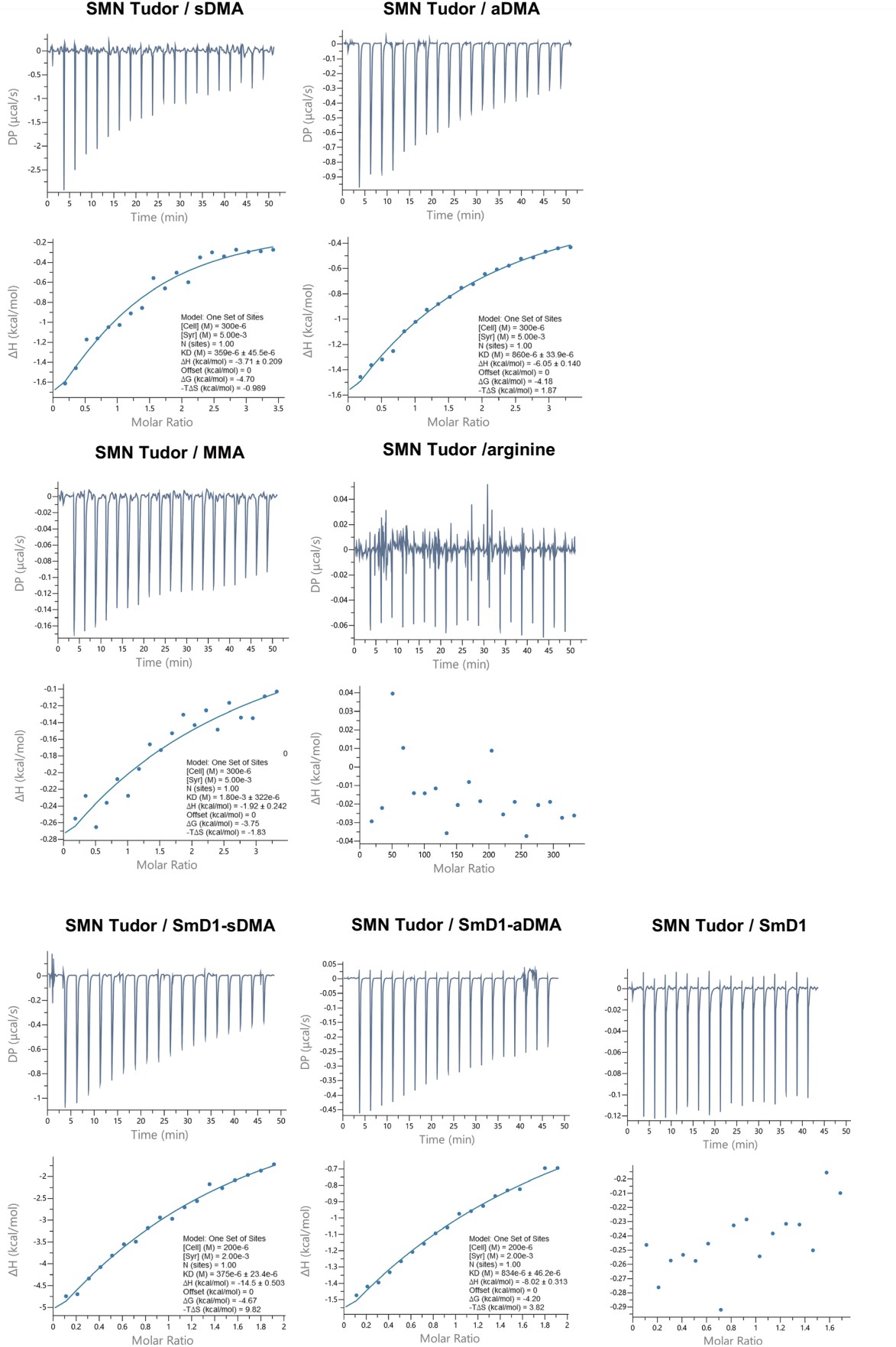

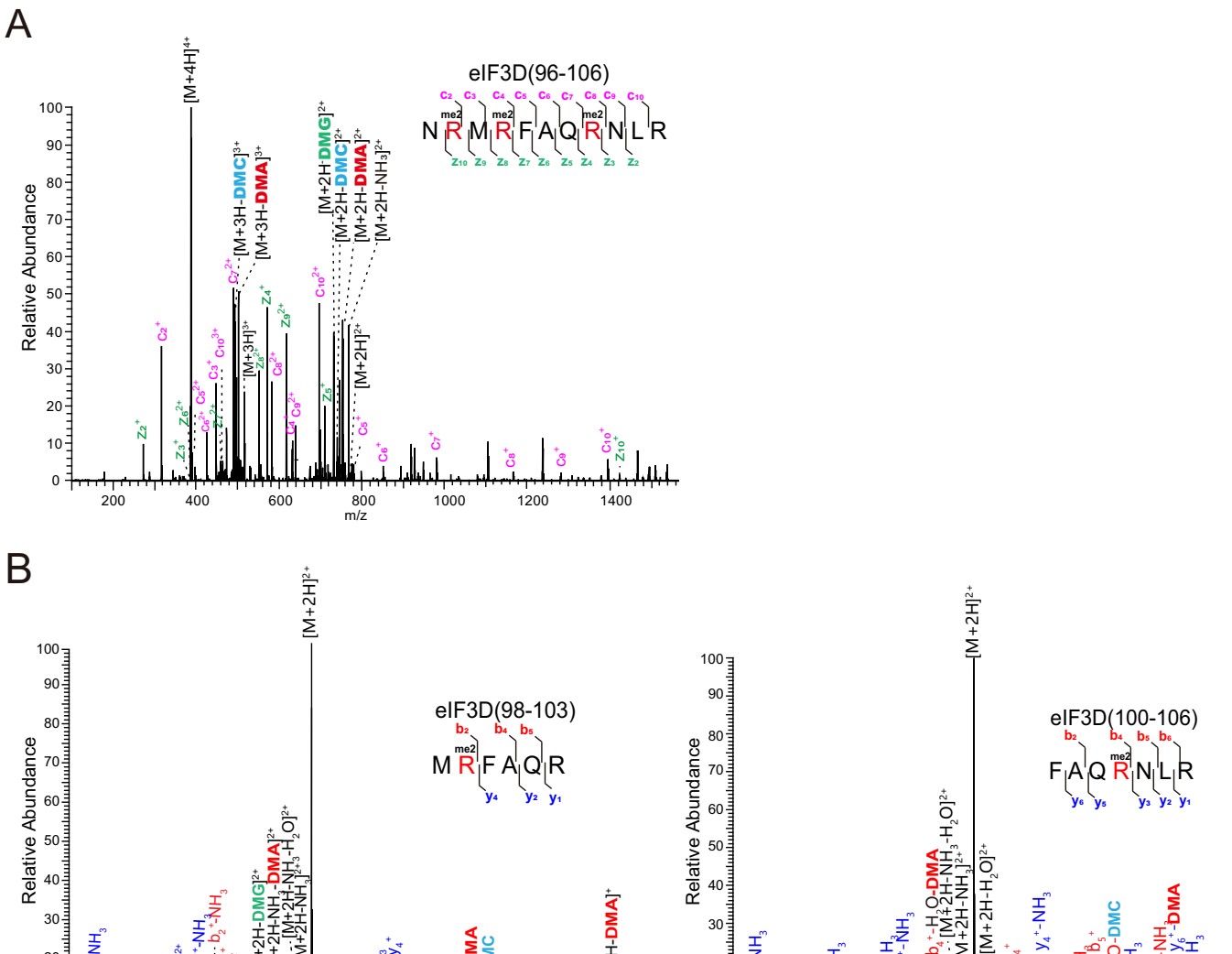

**Figure EV2. MS/MS spectrums of the dimethylated peptide from eIF3D.**

(A) MS/MS spectrum of the eIF3D peptide NR$_{me2}$MR$_{me2}$FAQR$_{me2}$NLR (96–106) identified by our molecular affinity approach. The spectrum was manually annotated with potential neutral losses. DMA, dimethylamine; DMC, dimethylcarbodiimide; DMG, dimethylguanidine. The abbreviations include their corresponding ions. (B) MS/MS spectrums of the eIF3D peptide MR$_{me2}$FAQR (98–103) and FAQR$_{me2}$NLR (100–106) from eIF3D by IP-MS. Plasmid constructs encoding the Full-length Flag-eIF3D proteins were transfected into HEK293T cells. eIF3D was immunoprecipitated with Flag agarose beads, digested with trypsin and subsequently subjected to mass spectrometry analysis.

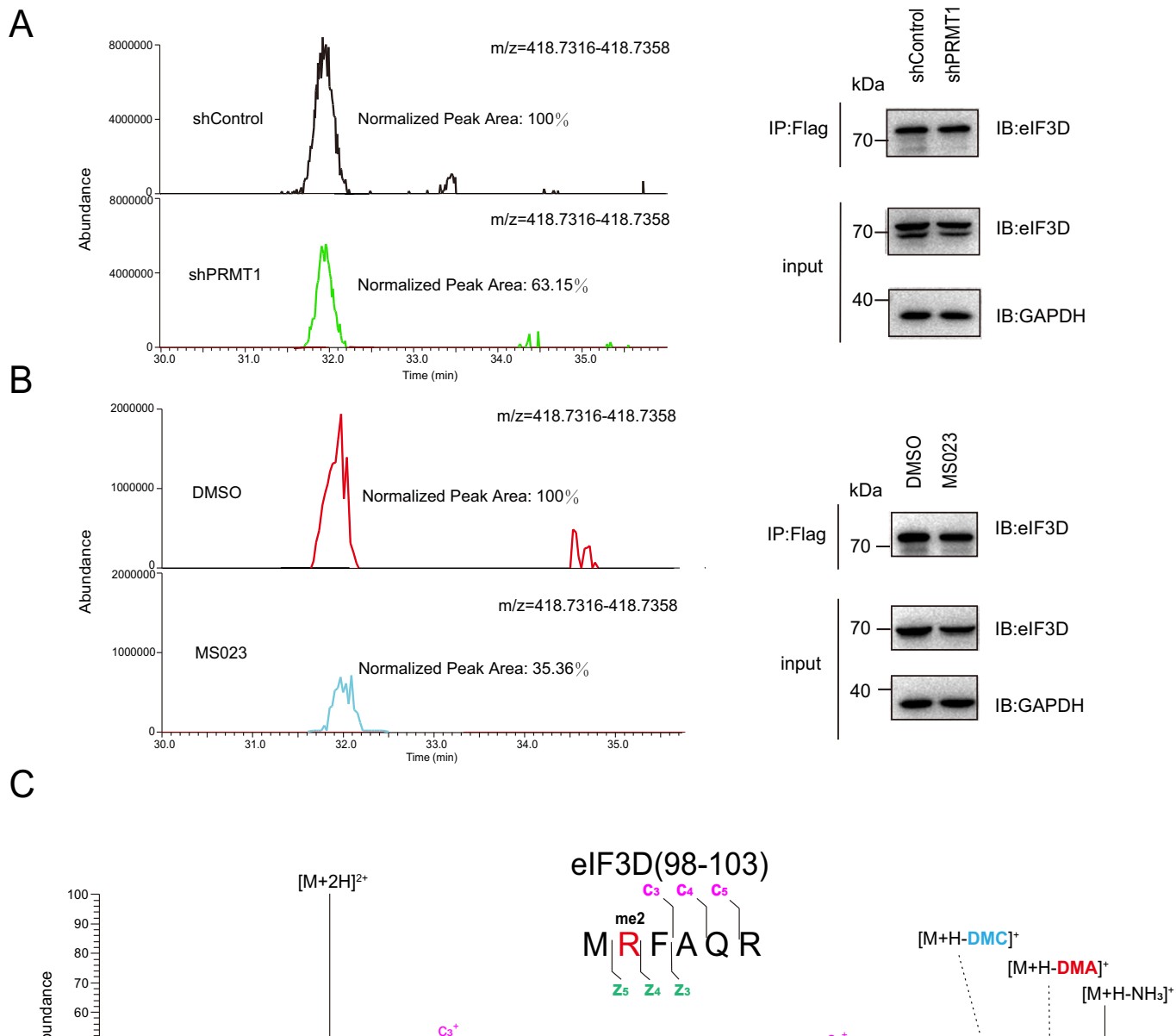

**Figure EV3.   IP-MS analysis of eIF3D R99 dimethylation in HEK293T cells.**

(A) Quantitative MS analysis of eIF3D peptide (MR$_{me2}$FAQR) with and without PRMT1 knockdown. (B) Quantitative MS analysis of eIF3D peptide (MR$_{me2}$FAQR) with pharmacological inhibition of Type I PRMTs. Inhibitor treatment was performed with HEK293T cells overexpressing wild-type eIF3D with 1 µM MS023 for 48 h. (C) MS/MS spectrum of eIF3D peptide (MR$_{me2}$FAQR) from the IP-MS experiment. The spectrum was manually annotated with potential neutral losses. DMA, dimethylamine; DMC, dimethylcarbodiimide. The abbreviations include their corresponding ions.

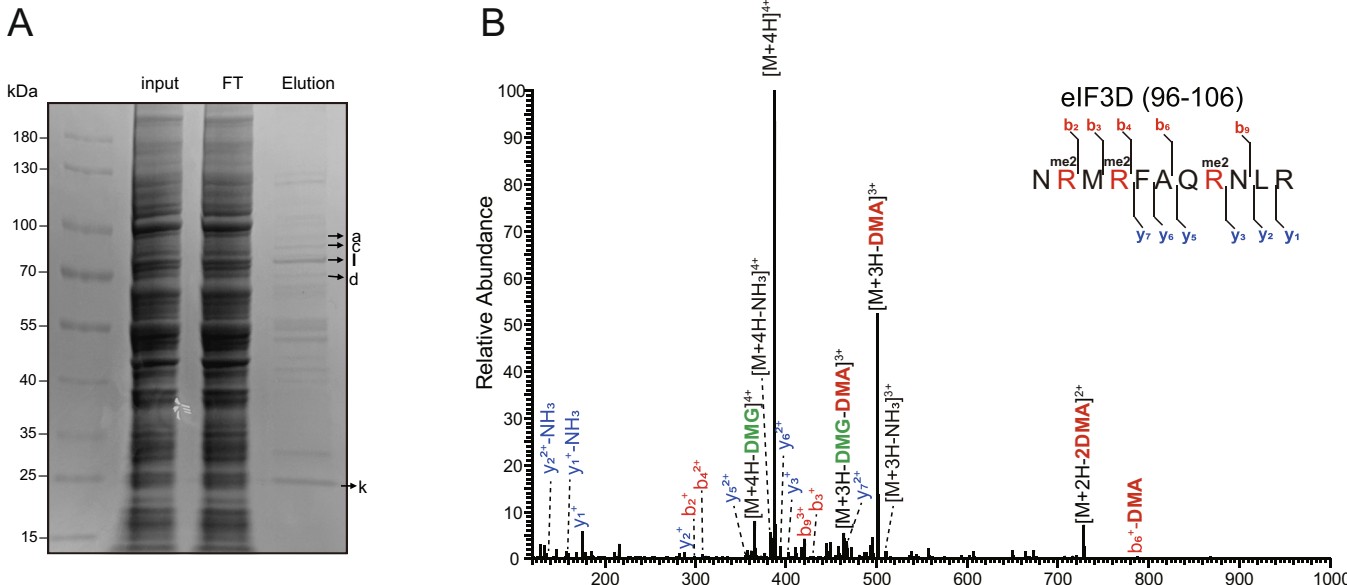

**Figure EV4. Analysis of eIF3D arginine methylation from eIF3 complex.**

(A) Coomassie stained SDS-PAGE gel showing purified eIF3 complex. (B) MS/MS spectrums of dimethylated eIF3D peptides NR$_{me2}$MR$_{me2}$FAQR$_{me2}$NLR (96–106) from immunoprecipated eIF3 complex. Plasmid constructs encoding the Full-length Flag-Myc-eIF3L proteins were transfected into HEK293T cells. eIF3L was immunoprecipitated with Flag agarose beads. The immunoprecipitated protein was divided into two aliquots: one portion was analyzed by Coomassie stain, and the other portion was digested with trypsin and subjected to mass spectrometry analysis. The spectrum was manually annotated with potential neutral losses. DMA, dimethylamine; DMG, dimethylguanidine. The abbreviations include their corresponding ions.

