## [Peer Review File · EMBO Reports]

Tudor-based Proteomic Strategy Pan-specifically Enriches and Identifies Protein Arginine Methylation

Lingzi Lu, Ting Li, Rou Zhang, Xiaoping Ye, Yutong Wang, Yixin Luo, Zilu Ye, Yang Mao, Yanqiu Yuan, Lingyu Sun, and Liang Qi

Corresponding author(s): Yanqiu Yuan (yuanyq8@mail.sysu.edu.cn), Yang Mao (maoyang3@mail.sysu.edu.cn), Zilu Ye (yzl@ism.pumc.edu.cn)

Review Timeline:

Submission Date:	13th Apr 25
Editorial Decision:	13th May 25
Revision Received:	13th Aug 25
Editorial Decision:	22nd Sep 25
Revision Received:	24th Sep 25
Accepted:	1st Oct 25

Editor: *Esther Schnapp*

Transaction Report:

Dear Prof. Yuan,

Thank you for the submission of your manuscript to EMBO reports. We have now received the full set of referee reports that is pasted below.

As you will see, the referees acknowledge that the findings are potentially interesting. However, they also have several suggestions for how the study could be improved. I think all suggestions are good and should be addressed. Please let me know in case you disagree and we can discuss the exact revision requirements further, also in a video chat, if you like.

I would thus like to invite you to revise your manuscript with the understanding that the referee concerns must be fully addressed and their suggestions taken on board. Please address all referee concerns in a complete point-by-point response. Acceptance of the manuscript will depend on a positive outcome of a second round of review. It is EMBO reports policy to allow a single round of major revision only and acceptance or rejection of the manuscript will therefore depend on the completeness of your responses included in the next, final version of the manuscript.

We realize that it is difficult to revise to a specific deadline. In the interest of protecting the conceptual advance provided by the work, we recommend a revision within 3 months (13th Aug 2025). Please discuss the revision progress ahead of this time with the editor if you require more time to complete the revisions.

- 1) A data availability section providing access to data deposited in public databases is missing. If you have not deposited any data, please add a sentence to the data availability section that explains that.
- 2) Your manuscript contains statistics and error bars based on $n=2$. Please use scatter blots in these cases. No statistics should be calculated if $n=2$.

5) a complete author checklist, which you can download from our author guidelines <<https://www.embopress.org/page/journal/14693178/authorguide>>. Please insert information in the checklist that is also reflected in the manuscript. The completed author checklist will also be part of the RPF.

6) Please note that all corresponding authors are required to supply an ORCID ID for their name upon submission of a revised manuscript (<<https://orcid.org/>>). Please find instructions on how to link your ORCID ID to your account in our manuscript tracking system in our Author guidelines <<https://www.embopress.org/page/journal/14693178/authorguide#authorshipguidelines>>

7) Before submitting your revision, primary datasets produced in this study need to be deposited in an appropriate public database (see <https://www.embopress.org/page/journal/14693178/authorguide#datadeposition>). Please remember to provide a reviewer password if the datasets are not yet public. The accession numbers and database should be listed in a formal "Data Availability" section placed after Materials & Method (see also <https://www.embopress.org/page/journal/14693178/authorguide#datadeposition>). Please note that the Data Availability Section is restricted to new primary data that are part of this study. * Note - All links should resolve to a page where the data can be accessed. *
If your study has not produced novel datasets, please mention this fact in the Data Availability Section.

12) All Materials and Methods need to be described in the main text using our 'Structured Methods' format, which is required for all research articles. According to this format, the Methods section includes a Reagents and Tools Table (listing key reagents, experimental models, software and relevant equipment and including their sources and relevant identifiers) followed by a Methods and Protocols section describing the methods using a step-by-step protocol format. The aim is to facilitate adoption of the methodologies across labs. More information on how to adhere to this format as well as a downloadable template (.docx) for the Reagents and Tools Table can be found in our author guidelines: <https://www.embopress.org/page/journal/14693178/authorguide#structuredmethods>.

An example of a Method paper with Structured Methods can be found here: <https://www.embopress.org/doi/full/10.1038/s44320-024-00037-6#sec-4>

I look forward to seeing a revised form of your manuscript when it is ready.

Referee #1:

In the manuscript titled "Tudor-based Proteomic Strategy Pan-specifically Enriches and Identifies Protein Arginine Methylation," the authors introduce a molecular affinity strategy leveraging the Tudor domain of SMN for the pan-specific enrichment of methylarginine-containing peptides, addressing a significant challenge in proteomics. The study demonstrates that this approach effectively identifies both mono- and dimethylated arginines in RGG/RG-rich and non-RG motifs, uncovering a total of 841 methylarginine sites, including 584 novel dimethylation sites. Notably, the authors identified asymmetric dimethylarginine (aDMA) at the R99 site of eIF3D, a key component of the eukaryotic translation initiation complex, and established that PRMT1 catalyzes this modification. This methylation event was shown to regulate both canonical and alternative cap-dependent translation initiation. Functional assays revealed that mutating R99 (R99K) reduces translation efficiency, underscoring the regulatory role of arginine methylation in protein synthesis. The strategy's broad specificity, low sample input requirement, and validation through heavy methyl SILAC (hmSILAC) and enzymatic assays highlight its potential for global proteomic profiling of arginine methylation. Additionally, the study provides mechanistic insights into the functional significance of arginine methylation, particularly in translation regulation, which may contribute to a deeper understanding of cellular processes and disease mechanisms. The findings also suggest that the Tudor domain-based approach complements existing methods, offering a flexible and cost-effective alternative for identifying previously unrecognized methylation sites and their biological functions.

Here is a summary of my main concerns regarding the manuscript:

The authors claim to have discovered 584 new dimethylarginine (DMA) methylation sites, which is an important finding. However, they only show this data in a chart (Figure 2E), which doesn't provide enough detail for others to verify the results. To make their work more solid, they should share a full list of the peptides they identified, including the peptide sequences, protein names, methylation locations, and false discovery rate (FDR) values, as these are key to assessing the accuracy of the findings. For the heavy methyl SILAC (hmSILAC) validation, they should also provide a detailed list of the peptides they confirmed, along with FDR values or other proof of reliability.

In addition to the UniProt database, which is often not up-to-date, the authors should compare their newly identified dimethylarginine (DMA) methylation sites with other comprehensive resources and published studies that have used antibody-based methods to profile arginine methylation sites. For example, studies like *J Am Soc Mass Spectrom.* 2023 (DOI: 10.1021/jasms.3c00154) and *Mol Cell Proteomics.* 2014 (DOI: 10.1074/mcp.O113.027870) have identified thousands of arginine methylation sites, far exceeding the limited number reported in UniProt (hundreds). Additionally, the authors should cross-reference their findings with databases like PhosphoSitePlus, which also catalogs post-translational modifications, including arginine methylation. By comparing their results with these extensive datasets, the authors can provide a more comprehensive and accurate validation of their novel DMA sites, ensuring their findings are robust and aligned with the latest research in the field. This approach would significantly enhance the credibility and impact of their study.

The isothermal titration calorimetry (ITC) results presented in the study are too simplistic and lack detailed information. The authors should include additional key parameters such as the stoichiometry (N), enthalpy change (ΔH), entropy change (ΔS), and Gibbs free energy change (ΔG).

Referee #2:

In this compelling manuscript, Lu et al show that the SMN tudor domain can be used to enrich for proteins containing dimethylarginine modifications. Given the difficulties in the field identifying all methylarginine substrates, this is an important advance and will allow novel science. Importantly, the authors also use their technology to identify the translation initiation factor eIF3D R99 and R103 methylation. They further validate the functional role of this modification by R to K mutation and show that cap-dependent translation of a luciferase reporter is reduced. While the technology cannot identify monomethylarginine-containing proteins, overall, this is a compelling study and should be published after addressing the following comments:

- Does SMN interaction depend at all on the peptide backbone or sequence? An ITC comparison of a short peptide (e.g. GRme2G?) with free arginine residues would be informative for the field and help validate the general utility of the technique
- How was the amino acid concentration determined for ITC? This should be reported in the methods, I did not see it listed anywhere
- In Fig 3F, R99me2a antibody needs better validation and support, perhaps with a peptide competition. The loss of signal in the PRMT1 knockdown is relatively modest
- Fig 4A input lanes, looks like in the shPRMT1 lane there is a reduction in eIF3D total protein which might obscure the R99me2 abundance blot, additional blots and/or quantification or proteomics would help. The FLAG IP eIF3D blot is also overexposed, perhaps obscuring reduced abundance.
- Note that after inspecting some of the published PTMScan studies supplemental sheets, eIF3D R99 and R103 methylation was previously found in <https://doi.org/10.1016/j.isci.2021.102971>

Referee #3:

In this manuscript, the Yuan lab reported their efforts on enriching methylarginine-containing peptides with Halo-tagged Tudor domain of SMN. It has been known that a subset of Tudor domains selectively recognize methylarginine peptides versus their cognate arginine peptides. Similar strategies to enrich dimethylarginine-containing peptides were shown to be successful with MBT domains. Here the authors demonstrated the feasibility to enrich several hundreds of target peptides from trypsin-digested HeLa cell lysate; conduct the motif analysis of the enriched peptides. Among these candidates, the authors primarily focused on aDMA at R99 of eIF3D and showed its dependence on PRMT1's activity and studied its potential role on translational initiation. This work has its merits in terms of the technology development and biological discovery with the central theme on protein arginine methylation. Meanwhile, the reviewer brought out the following concerns for the authors to address upon revision:

- (1) It has been known that the Tudor domain of SMN strongly prefers the peptides containing sDMA versus aDMA and MMA regarding the affinity assays with peptides and structural evidence. However, it is surprising that the selectivity for sDMA and aDMA is only 1.3-fold. It is not clear that such discrepancy is due to the intrinsic character in recognition of sDMA and aDMA or the perturbation in the context of the particular construct, Halo-tag, or C-6xHis tag or their combination. Can authors test the reported peptides to clarify this point?
- (2) In the manuscript, the authors claimed that their Tudor fusion protein doesn't bind Arg and the mutant doesn't bind sDMA. However, such experiments were conducted with Arg without the context of peptide sequences. Given the K_d values are close to the upper limitation of the assay condition, it is unclear whether it just weakly binds but beyond their assay threshold. In theory, "n.b" should be written as "> xxx mM" derived on the basis of their assay conditions.
- (3) It is also remarkable that Y127F doesn't bind sDMA. Is there other evidence to show the point mutation of Y127F rather than Y127A had such a dramatic effect?
- (4) In Fig 2E, the authors made comparison of their dataset with Uniport. However, it is remarkable about the limited overlap. The authors need to discuss and justify this difference.
- (5) In addition, the authors generated their peptides with trypsin digestion cutting off K/R sites. However, many methylarginine sites are embedded within K/R-rich regions? Will the step limit the number of the revealed peptides as well? More analysis in this regard is required.
- (6) Regarding the function of R99, the key experiments were conducted by comparing the outcomes of wt versus R99K variants. However, such experiments won't eliminate the role of R99 itself versus R99K mutant. Though indefinitely, such experiments should be conducted in the presence of genetic/pharmacological perturbation of PRMT1.
- (7) Although PRMT5 showed no activity on the eIF3D peptide in vitro. It is not clear whether the R99 site can be modified by PRMT5 in the presence or absence of PRMT1 in a cellular context. Is there any way to show the sole aDMA of R99 in a cellular context or in the presence of PRMT1 inhibitors?

Referee #1:

In the manuscript titled "Tudor-based Proteomic Strategy Pan-specifically Enriches and Identifies Protein Arginine Methylation," the authors introduce a molecular affinity strategy leveraging the Tudor domain of SMN for the pan-specific enrichment of methylarginine-containing peptides, addressing a significant challenge in proteomics. The study demonstrates that this approach effectively identifies both mono- and dimethylated arginines in RGG/RG-rich and non-RG motifs, uncovering a total of 841 methylarginine sites, including 584 novel dimethylation sites. Notably, the authors identified asymmetric dimethylarginine (aDMA) at the R99 site of eIF3D, a key component of the eukaryotic translation initiation complex, and established that PRMT1 catalyzes this modification. This methylation event was shown to regulate both canonical and alternative cap-dependent translation initiation. Functional assays revealed that mutating R99 (R99K) reduces translation efficiency, underscoring the regulatory role of arginine methylation in protein synthesis. The strategy's broad specificity, low sample input requirement, and validation through heavy methyl SILAC (hmSILAC) and enzymatic assays highlight its potential for global proteomic profiling of arginine methylation. Additionally, the study provides mechanistic insights into the functional significance of arginine methylation, particularly in translation regulation, which may contribute to a deeper understanding of cellular processes and disease mechanisms. The findings also suggest that the Tudor domain-based approach complements existing methods, offering a flexible and cost-effective alternative for identifying previously unrecognized methylation sites and their biological functions.

Here is a summary of my main concerns regarding the manuscript:

1. The authors claim to have discovered 584 new dimethylarginine (DMA) methylation sites, which is an important finding. However, they only show this data in a chart (Figure 2E), which doesn't provide enough detail for others to verify the results. To make their work more solid, they should share a full list of the peptides they identified, including the peptide sequences, protein names, methylation locations, and false discovery rate (FDR) values, as these are key to assessing the accuracy of the findings. For the heavy methyl SILAC (hmSILAC) validation, they should also provide a detailed list of the peptides they confirmed, along with FDR values or other proof of reliability.

Response: We appreciate the reviewer's constructive feedback. In response, the revised manuscript now incorporates a supplementary data file (Table EV1.xlsx) containing comprehensive lists of all methylarginine peptides identified by our method and those validated by hmSILAC in this study. This dataset is accompanied by detailed annotations of their respective molecular characteristics, including sequences, intensities, methylation sites, and parameters to assess reliability (Percolator q-Value and PEP).

2. In addition to the UniProt database, which is often not up-to-date, the authors should

compare their newly identified dimethylarginine (DMA) methylation sites with other comprehensive resources and published studies that have used antibody-based methods to profile arginine methylation sites. For example, studies like *J Am Soc Mass Spectrom.* 2023 (DOI: 10.1021/jasms.3c00154) and *Mol Cell Proteomics.* 2014 (DOI: 10.1074/mcp.O113.027870) have identified thousands of arginine methylation sites, far exceeding the limited number reported in UniProt (hundreds). Additionally, the authors should cross-reference their findings with databases like PhosphoSitePlus, which also catalogs post-translational modifications, including arginine methylation. By comparing their results with these extensive datasets, the authors can provide a more comprehensive and accurate validation of their novel DMA sites, ensuring their findings are robust and aligned with the latest research in the field. This approach would significantly enhance the credibility and impact of their study.

Response: In accordance with the reviewer's recommendation, we first compared the number of dimethylarginine (DMA) sites identified from this study with a prior large-scale investigation using aDMA-specific antibodies as the reviewer suggested (PMID: 37463068; 24129315) as well as recent methylarginine profiling studies using sDMA-specific antibodies (PMID: 38272855; 32468700). The number of DMA sites identified in our study was comparable, if not higher, than previously reported numbers from a single study (The aDMA antibodies were reported to identify between 300 and 500 sites in HCT116 cells and the sDMA antibodies were reported to identify between 100 and 250 sites in human cells). We also compared the list of DMA sites identified in our study to the PhosphoSitePlus database (see below and Figure 2E of the revised manuscript). Of the 743 DMA sites, 391 were previously identified and 352 were novel. In addition, 46 proteins were newly found to be modified with DMA. Lastly, a comparison between the sites identified in our study with those reported in Uniprot showed that our method could identify both aDMA and sDMA, reflecting the ability of SMN to recognize both DMA forms (Appendix Figure S4). We have revised the manuscript to include these comparisons we've made.

Figure 2E. Overlaps of DMA sites and DMA-modified proteins between this study and PhosphoSitePlus database.

3. The isothermal titration calorimetry (ITC) results presented in the study are too simplistic and lack detailed information. The authors should include additional key parameters such as the stoichiometry (N), enthalpy change (ΔH), entropy change (ΔS), and Gibbs free energy change (ΔG).

Response: In response to the reviewer's feedback, the thermodynamic parameters derived ITC experiments are now presented in Table 1, with a notation that all data fitting operations employed a fixed stoichiometry (N) of 1 and a 1:1 binding mode. To improve clarity, we have now integrated these parameters as well as the stoichiometry (N) directly into their corresponding ITC titration curves, which are provided in Figure EV1 of the revised manuscript.

Table 1. Thermodynamic parameters for binding between SMN Tudor protein and various methylated arginine forms. A fixed stoichiometry (N) of 1 and a 1:1 binding mode was employed for all data fitting.

Ligand	K_D (mM)	ΔH (kcal/mol)	$-T\Delta S$ (kcal/mol)	ΔG (kcal/mol)
sDMA	0.359 ± 0.045	-3.71 ± 0.209	-0.99	-4.70
aDMA	0.860 ± 0.034	-6.05 ± 0.140	1.87	-4.18
MMA	1.800 ± 0.322	-1.92 ± 0.242	-1.83	-3.75
Arginine	> 5 mM	n.c.	n.c.	n.c.
SmD1-sDMA	0.375 ± 0.023	-14.50 ± 0.503	9.82	-4.67
SmD1-aDMA	0.834 ± 0.046	-8.02 ± 0.313	3.82	-4.20
SmD1	> 2 mM	n.c.	n.c.	n.c.

n.c.: not calculatable.

Referee #2:

In this compelling manuscript, Lu et al show that the SMN tudor domain can be used to enrich for proteins containing dimethylarginine modifications. Given the difficulties in the field identifying all methylarginine substrates, this is an important advance and will allow novel science. Importantly, the authors also use their technology to identify the translation initiation factor eIF3D R99 and R103 methylation. They further validate the functional role of this modification by R to K mutation and show that cap-dependent translation of a luciferase reporter is reduced. While the technology cannot identify monomethylarginine-containing proteins, overall, this is a compelling study and should be published after addressing the

following comments:

1. Does SMN interaction depend at all on the peptide backbone or sequence? An ITC comparison of a short peptide (e.g. GRme2G?) with free arginine residues would be informative for the field and help validate the general utility of the technique

Response: We appreciate the reviewer's insightful comment regarding the role of flanking residues in methylarginine-Tudor domain interactions. As highlighted in a study by Tripsianes et al. (PMID: 22101937), the binding affinity of the SMN Tudor domain for a pentapeptide containing a single symmetric dimethylarginine (sDMA) residue ($K_D=0.507$) was nearly equivalent to that of free sDMA ($K_D=0.476$). NMR titrations confirmed negligible contributions from adjacent residues. To address the reviewer's concern and evaluate this phenomenon in our system, we synthesized heptapeptides derived from the Sm D1 protein, each containing no modification (SmD1), one aDMA (SmD1-aDMA), or one sDMA modification (SmD1-sDMA). We performed ITC experiments to compare the binding thermodynamics between these peptides and the SMN Tudor. As demonstrated in the Table 1 (see below), our results corroborate Tripsianes et al.'s findings that flanking sequences exert minimal influence on SMN Tudor's binding affinities toward sDMA or aDMA. This analysis not only reinforces the broader mechanistic principle that SMN Tudor domain specificity is driven predominantly by methyl-arginine recognition rather than contextual sequence features, but also sets a solid foundation for our enrichment method. We have included the new data in Table 1 and Figure EV1 of the revised manuscript.

Table 1. Thermodynamic parameters for binding between SMN Tudor protein and various methylated arginine forms. A fixed stoichiometry (N) of 1 and a 1:1 binding mode was employed for all data fitting.

Ligand	K_D (mM)	ΔH (kcal/mol)	$-T\Delta S$ (kcal/mol)	ΔG (kcal/mol)
sDMA	0.359 ± 0.045	-3.71 ± 0.209	-0.99	-4.70
aDMA	0.860 ± 0.034	-6.05 ± 0.140	1.87	-4.18
MMA	1.800 ± 0.322	-1.92 ± 0.242	-1.83	-3.75
Arginine	> 5 mM	n.c.	n.c.	n.c.
SmD1-sDMA	0.375 ± 0.023	-14.50 ± 0.503	9.82	-4.67
SmD1-aDMA	0.834 ± 0.046	-8.02 ± 0.313	3.82	-4.20
SmD1	> 2 mM	n.c.	n.c.	n.c.

n.c.: not calculatable.

Figure EV1. ITC titrations of SMN Tudor protein with Smd1-derived heptapeptides or

methylarginines. The lines represent data fitting curves using 1:1 binding model at a fixed stoichiometry (N) of 1. The peptides used were SmD1 (AGRGRGR), SmD1-sDMA (AGR_{me2s}GRGR), SmD1 aDMA (AGR_{me2a}GRGR).

2. How was the amino acid concentration determined for ITC? This should be reported in the methods, I did not see it listed anywhere

Response: The following clarification has been added to the Methods section regarding amino acid quantification: L-Arginine, symmetric dimethylarginine (sDMA), and asymmetric dimethylarginine (aDMA) were purchased from MedChemExpress (Darmstadt, Germany) with purity exceeding 98.0% (HPLC-verified). Stock solutions were prepared gravimetrically in phosphate buffered saline (PBS, pH 7.4) and stored at -80°C until use.

3. In Fig 3F, R99me2a antibody needs better validation and support, perhaps with a peptide competition. The loss of signal in the PRMT1 knockdown is relatively modest.

Response: We appreciate the reviewer's comment. The IP western experiment with PRMT1 knockdown using R99me2a antibody was repeated three times and dimethylated eIF3D was quantified by densitometry and normalized against eIF3D (See below). Statistical analysis showed a significant reduction in R99me2a signal upon PRMT1 knockdown. We included the statistical analysis in Figure 4A of the revised manuscript.

Figure 4A. Immunoblot and statistical analysis of eIF3D R99 aDMA modification in HEK293T cells with and without PRMT1 knockdown. Data are presented as mean±SD; *P = 0.0161 by unpaired t test; n = 3.

To address the reviewer's concern and to cross validate the functional role of PRMT1 in eIF3D R99 methylation, we performed IP-MS with and without PRMT1 knockdown. eIF3D was overexpressed in HEK293 cells, pulled down by anti-flag antibody in parallel, trypsin

digested and submitted to LC-MS analysis. Quantification of mass spectrometry peaks corresponding to the dimethylated R99-containing peptide (MR_{me2}FAQR) revealed an approximately 40% reduction in peak intensity following PRMT1 knockdown (Figure EV3). This observation corroborates previous immunoblot findings, indicating PRMT1 as the primary enzyme responsible for R99 dimethylation. However, the persistence of residual dimethylation suggests the involvement of redundant enzymatic activities targeting R99 in the absence of PRMT1. To further investigate this hypothesis, we pharmacologically inhibited type I PRMTs using MS023, followed by IP-MS analysis of eIF3D. A more substantial reduction of 65% in the peak intensity of dimethylated R99 was observed, indicating potential contributions from other PRMTs to R99 dimethylation. Nevertheless, the specific identity of these contributing PRMTs remains to be elucidated. These new findings and their implications have been incorporated into the revised manuscript.

Figure EV3. IP-MS analysis of eIF3D R99 dimethylation in HEK293T cells. (A) Quantitative MS analysis of eIF3D peptide (MR_{me2}FAQR) with and without PRMT1 knockdown. (B) Quantitative MS analysis of eIF3D peptide (MR_{me2}FAQR) with pharmacological inhibition of Type I PRMTs. Inhibitor treatment was performed with HEK293T cells overexpressing wild-type eIF3D with 1 μM MS023 for 48 hours. (C) MS/MS spectrum of eIF3D peptide (MR_{me2}FAQR) from the IP-MS experiment. The spectrum was manually annotated with potential neutral losses. Abbreviations: DMA, dimethylamine; DMC, dimethylcarbodiimide. The abbreviations include their corresponding ions.

4. Fig 4A input lanes, looks like in the shPRMT1 lane there is a reduction in eIF3D total protein which might obscure the R99me2 abundance blot, additional blots and/or quantification or proteomics would help. The FLAG IP eIF3D blot is also overexposed, perhaps obscuring reduced abundance.

Response: Please refer to our response to Question 3.

5. Note that after inspecting some of the published PTMScan studies supplemental sheets, eIF3D R99 and R103 methylation was previously found in <https://doi.org/10.1016/j.isci.2021.102971>

Response: We appreciate the reviewer's attention to this detail. While methylation at eIF3D arginine residues R99 and R103 was indeed identified in prior studies, e.g. PMID: 34505004, these sites were not validated by hmSILAC or quantitative analysis following pharmacological inhibition of PRMTs (fold change <1.5; p > 0.05). False positive identification can occur in LC-MS analysis of methylation sites due to the isobaric nature of methylation to several amino acid substitutions, therefore these sites in eIF3D were not considered validated methylation sites. To accurately reflect previous findings and our study, we revised the manuscript in line 193-194 as follows: Dimethylation at R99 and R103 in the eukaryotic initiation factor 3D protein is among the newly validated modification sites by our molecular affinity strategy.

Referee #3:

In this manuscript, the Yuan lab reported their efforts on enriching methylarginine-containing peptides with Halo-tagged Tudor domain of SMN. It has been known that a subset of Tudor domains selectively recognizes methylarginine peptides versus their cognate arginine peptides. Similar strategies to enrich dimethylarginine-containing peptides were shown to be successful with MBT domains. Here the authors demonstrated the feasibility to enrich several hundreds of target peptides from trypsin-digested HeLa cell lysate; conduct the motif analysis of the enriched peptides. Among these candidates, the

authors primarily focused on aDMA at R99 of eIF3D and showed its dependence on PRMT1's activity and studied its potential role on translational initiation. This work has its merits in terms of the technology development and biological discovery with the central theme on protein arginine methylation. Meanwhile, the reviewer brought out the following concerns for the authors to address upon revision:

(1) It has been known that the Tudor domain of SMN strongly prefers the peptides containing sDMA versus aDMA and MMA regarding the affinity assays with peptides and structural evidence. However, it is surprising that the selectivity for sDMA and aDMA is only 1.3-fold. It is not clear that such discrepancy is due to the intrinsic character in recognition of sDMA and aDMA or the perturbation in the context of the particular construct, Halo-tag, or C-6xHis tag or their combination. Can authors test the reported peptides to clarify this point?

Response: Thank you for pointing this out. The SMN Tudor domain was reported to preferentially bind to sDMA over its aDMA, with reported dissociation constants of 0.476 mM for sDMA and 1.025 mM for aDMA, yielding a ~2-fold affinity difference (PMID: 22101937). To make a better comparison, we re-performed our ITC experiments using the same concentration of ligands for all titrations, which yielded K_D values of 0.359 mM for sDMA and 0.860 mM for aDMA, corresponding to a ~2-fold difference, similar to the reported values. In addition, to investigate if flanking residues in the peptide affect binding preferences, we performed ITC between SMN Tudor domain and a SmD1 derived pentapeptide containing either sDMA or aDMA. As shown in the table below, their absolute affinity and preference for sDMA modification is similar to what we obtained with free amino acids. While the absolute affinities differ slightly between our experiments and prior studies, which could be due to differences in Tudor domain boundaries (literature: residues 84–147 vs. our construct: 80-167) or buffer composition (literature: 20 mM sodium phosphate (pH 6.5), 50 mM NaCl vs. our study: phosphate buffered saline (PBS, pH 7.4)), both studies affirm the Tudor domain's specific binding to DMA, with preferential recognition of sDMA.

Table 1. Thermodynamic parameters for binding between wild type SMN Tudor protein and methylarginines. A fixed stoichiometry (N) of 1 and a 1:1 binding mode was employed for all data fitting.

Ligand	K_D (mM)	ΔH (kcal/mol)	$-T\Delta S$ (kcal/mol)	ΔG (kcal/mol)
sDMA	0.359 ± 0.045	-3.71 ± 0.209	-0.99	-4.70
aDMA	0.860 ± 0.034	-6.05 ± 0.140	1.87	-4.18
MMA	1.800 ± 0.322	-1.92 ± 0.242	-1.83	-3.75
Arginine	> 5 mM	n.c.	n.c.	n.c.

SmD1-sDMA	0.375 ± 0.023	-14.50 ± 0.503	9.82	-4.67
SmD1-aDMA	0.834 ± 0.046	-8.02 ± 0.313	3.82	-4.20
SmD1	> 2 mM	n.c.	n.c.	n.c.

n.c.: not calculatable.

(2) In the manuscript, the authors claimed that their Tudor fusion protein doesn't bind Arg and the mutant doesn't bind sDMA. However, such experiments were conducted with Arg without the context of peptide sequences. Given the K_d values are close to the upper limitation of the assay condition, it is unclear whether it just weakly binds but beyond their assay threshold. In theory, "n.b" should be written as "> xxx mM" derived on the basis of their assay conditions.

Response: Thank you for pointing this out. We performed additional ITC experiments between SMN Tudor domain and a SmD1 derived pentapeptide containing either no methylation, one sDMA or aDMA. As shown in the table above, their absolute affinity and preference for sDMA modification is similar to what we obtained with free amino acids. We used a ligand concentration of 2 mM for the SmD1 peptide and 5mM for Arginine, therefore we changed n.b. to > 2 mM and > 5mM, respectively, to reflect our assay condition.

(3) It is also remarkable that Y127F doesn't bind sDMA. Is there other evidence to show the point mutation of Y127F rather than Y127A had such a dramatic effect?

Response: Thank you for pointing this out. Upon re-examining our construct sequence, we found that Y127F was a typo in the original manuscript. It should be Y127L (see below for the sequencing result). The ITC result with Y127L is consistent with literature findings (PMID: 22101937). To further validate the binding specificity of the Tudor domain, we conducted ITC analyses using other aromatic cage mutants, including W102F and N132E. None of the mutants exhibited significant binding to SMN Tudor at 5 mM sDMA concentration (see below), reinforcing the specificity underlying SMN Tudor-methylarginine interactions. The additional ITC data is included in Appendix Figure S1 of the revised manuscript.

DNA sequencing result of SMN Tudor Y127L mutant:

SMN Tudor / sDMA

Y127L mutant / sDMA

N132E mutant / sDMA

W102F mutant / sDMA

Appendix Figure S1. ITC titrations of wild type or mutant SMN Tudor with sDMA.

(4) In Fig 2E, the authors made comparison of their dataset with Uniport. However, it is remarkable about the limited overlap. The authors need to discuss and justify this difference.

Response: We appreciate the reviewer's comment. Due to the delays in updates within UniProt, there is a possibility of underestimating the overlap of sites between our study and prior studies. To address this limitation, we utilized an alternative database, PhosphoSitePlus, which offers a more comprehensive and timely resource for post-translational modifications (PTMs), facilitating a more accurate comparison in our analysis. Of the 743 DMA sites from our study, 391 were previously identified and 352 were novel (see below). In addition, 46 proteins were newly found to be modified with DMA. We have replaced Figure 2E with the current comparison.

Figure 2E. Overlaps of DMA sites and DMA-modified proteins between this study and PhosphoSitePlus database.

(5) In addition, the authors generated their peptides with trypsin digestion cutting off K/R sites. However, many methylarginine sites are embedded within K/R-rich regions? Will the step limit the number of the revealed peptides as well? More analysis in this regard is required.

Response: We appreciate the reviewer's insightful critique regarding the potential impact of proteolytic miscleavage on methylarginine peptide identification. While trypsin cleavage is less efficient at methylated arginine sites (PMID: 19195997), potentially generating long peptides outside the optimal mass range for mass spectrometry detection, the presence of multiple methylarginine residues within a single peptide may increase its chance of being enriched by our affinity resin, offsetting potential losses in identifications due to suboptimal peptide lengths. As empirically demonstrated in Figure 2A, >85% of enriched peptides contained two or more methylarginine residues, a trend consistent with prior large-scale methylarginine studies (PMID: 37463068;33782401;34505004).

Ulilysin is an alternative protease that could be used for proteomic profiling of cellular arginine methylation (PMID: 25419962). It specifically cleaves at the N-terminal of arginine or lysine residues regardless of methylation status, thus generating shorter

methylarginine-containing peptides. During our initial method development, we did comparative analysis using either ulilysin or trypsin, and found that the number of arginine methylation sites identified through Trypsin digestion is slightly higher than Ulilysin (see below), which could be the result of higher enrichment of multimethylarginine containing peptides. Therefore, we chose Trypsin as the digestion enzyme for our Tudor-affinity enrichment.

Comparison of methylated peptides and sites identified from Hela cells treated with ulilysin or trypsin digestion by molecular affinity enrichment.

(6) Regarding the function of R99, the key experiments were conducted by comparing the outcomes of wt versus R99K variants. However, such experiments won't eliminate the role of R99 itself versus R99K mutant. Though indefinitely, such experiments should be conducted in the presence of genetic/pharmacological perturbation of PRMT1.

Response: We appreciate the reviewer's insightful comment. While the comparison between wild-type and R99K mutant constructs offers valuable information regarding the functional role of arginine methylation, we recognize the inherent limitation posed by the distinct biochemical properties of arginine and lysine. Our initial reluctance to perform PRMT1 knockdown or inhibition experiments stemmed from PRMT1's broad substrate specificity and its critical involvement in numerous cellular activities; its perturbation can lead to widespread changes in cellular transcription and translation, potentially obscuring specific effects.

Nevertheless, to directly address the reviewer's concern, we performed the luciferase reporter assay in the presence or absence of MS023, a pharmacological inhibitor that effectively reduced R99 dimethylation in our system. As depicted below, while the cells harboring R99K mutation in eIF3D consistently exhibited reduced luciferase signals, indicating decreased canonical translation initiation, the introduction of MS023 treatment eliminated the significant difference between the wild-type and R99K mutant, suggesting

that the observed reduction in the R99K mutant was likely attributable to the absence of arginine methylation. However, the interpretation of these results is complicated by a substantial and unexpected increase in the overall luciferase signal in cells treated with MS023. Given that understanding the upregulation of translation following global aDMA inhibition requires substantial further investigation, we believe these data are not yet suitable for inclusion in the manuscript.

We note that RK mutants are commonly utilized in the literature to demonstrate the functional significance of arginine methylation (e.g., [PMID: 33459381 (EMBO 2021) Figure 5,6,7]), possibly for reasons similar to our hesitation regarding PRMT1 perturbation. Acknowledging these experimental considerations, we have added the limitation of our method in the discussion section of the revised manuscript.

Luciferase activity in HEK293T cells stably expressing wild-type eIF3D and R99K with 1 μ M MS023 for 48 hours, using mRNAs carrying a canonical 5'UTR. Data are presented as mean \pm SD; ***P = 0.0001 and ****P < 0.0001 by unpaired t test; n = 4.

(7) Although PRMT5 showed no activity on the eIF3D peptide in vitro. It is not clear whether the R99 site can be modified by PRMT5 in the presence or absence of PRMT1 in a cellular context. Is there any way to show the sole aDMA of R99 in a cellular context or in the presence of PRMT1 inhibitors?

Response: Thank you for pointing this out. The initial indication that R99 was modified by aDMA stemmed from the MS² spectrum of the NR_{me2}MR_{me2}FAQR_{me2}NLR peptide during the proteomic analysis of HeLa cell (see below: panel A) This spectrum exhibited a characteristic -DMA ion, consistent with dimethylamine neutral loss upon aDMA fragmentation. Crucially, the absence of a diagnostic ion corresponding to monomethylamine neutral loss, which is typical of sDMA fragmentation, suggested that R99 was aDMA-modified and that

PRMT5-mediated sDMA at this site was unlikely in the cellular context. The neutral loss from dimethylated arginine peptides is a well-established mass spectrometric signature and is widely utilized for confirming the identity of two DMA forms (PMID: 15095356; 30940768; 38272855). The observation of a similar MS² spectrum for the MR_{me2}FAQR peptide from eIF3D overexpressed and immunoprecipitated from HEK293 cells (see below: panel B) further corroborated the aDMA modification of R99 within the cellular environment.

To address the reviewer's concern if PRMT5 could modify R99 in the absence of PRMT1 activity, we conducted immunoprecipitation-mass spectrometry (IP-MS) analyses following PRMT1 knockdown and pharmacological inhibition with MS023. However, a substantial reduction in the mass spectrometry signal for the MR_{me2}FAQR peptide (other R99_{me2}-containing peptides as well) was observed under both PRMT1 knockdown and MS023 treatment conditions, preventing us from obtaining a good quality MS² spectrum. Although the results suggest that R99 dimethylation is predominantly catalyzed by Type I PRMTs, the potential for PRMT5 to contribute to this modification in the absence of Type I PRMT activity cannot be conclusively dismissed. A discussion on this matter is included in the revised manuscript.

A

B

A. MS/MS spectrum of the eIF3D peptide (NR_{me2}MR_{me2}FAQR_{me2}NLR) from the proteomic analysis of HeLa cell by our molecular affinity approach. B. MS/MS spectrums of the eIF3D peptide MR_{me2}FAQR from the IP-MS experiment. The spectra were manually annotated with potential neutral losses. Abbreviations: DMA, dimethylamine; DMC, dimethylcarbodiimide; DMG, dimethylguanidine. The abbreviations include their corresponding ions.

Dear Prof. Yuan

Thank you for the submission of your revised manuscript. We have now received the enclosed reports from the referees and I am happy to say that all support its publication now. Only a few editorial requests will need to be addressed before we can proceed with the official acceptance of your manuscript.

- Your study has 5 main figures and will thus be published as a short report with combined results and discussion sections. Can you please combine both sections?
- The author credits need to be removed from the ms file. All credits need to be entered during online ms submission.
- The REFERENCE format needs to be alphabetical, not numerical; et al needs to be used after 10 author names. Please use the EMBO reports reference style.
- The Excel file with several sheets that is titled Table EV1 should be called Dataset EV1. You can either have
 - a) one Dataset EV1 file with multiple sheets/tabs but then the sheet titles (legends) should NOT be Supplementary Table 2.1, 2.2., etc. maybe just have the title and legend without any numbers OR
 - b) each sheet/tab needs to be uploaded separately as Dataset EV1, Dataset EV2, etc.

The Datasets should be called out in the text as such ("Supplemental" or "Supplementary" should not be used).

- The APPENDIX FILE is OK, but the "Supporting Information for" should be replaced with "Appendix for". The Appendix table should be part of the Reagents and Tools table. The Reagents and Tools table that is in the ms needs to be removed, we only need it as a separate file.
- The synopsis image you sent has too small text at the final image size. I attach it here for your information. Please send us a new synopsis image with bigger text font. The image can be higher but must be 550 pixels wide.
- I slightly modified your short summary and bullet points. Please let me know whether you agree with this:

This study establishes a molecular affinity approach for proteomic profiling of arginine methylation, showing broad specificity and high efficiency. It also reveals a regulatory role of arginine methylation in eukaryotic protein translation initiation.

- A molecular affinity strategy based on the SMN Tudor domain shows broad specificity and high efficiency for global proteomic profiling of arginine methylation.
 - eIF3D is identified as a target of asymmetric dimethylarginine modification within its RNA-binding region, affecting cap-dependent translation initiation.
 - The molecular affinity strategy provides an alternative tool for investigating the functions of arginine methylation.
- Please add the specific URLs for some of the Source Data (SD) to the Data Availability Section. The specific URL for the PXD053390 dataset also needs to be provided in the data availability section. Also, 1 SD folder per main figure needs to be uploaded.

- Material and Methods should be just Methods.

Figure Legends - Comments

- Please note that the exact p values are not provided in the legend of figure 5C. Please provide exact values as reasonable.
- Please indicate the statistical test used for data analysis in the legend of figure 2D.

I would like to suggest some changes to the abstract that needs to be written in present tense. Please let me know whether you agree with the following:

Protein arginine methylation is an important post-translational modification (PTM) in eukaryotes, regulating a variety of biological processes. Proteomic profiling of arginine methylation has advanced our understanding of its roles in biology and disease. However, pan-specific enrichment of methylarginine-containing peptides remains challenging. Here we report a molecular affinity strategy based on the Tudor domain of SMN, a naturally occurring methylarginine reader protein, for comprehensive proteomic profiling of cellular arginine methylation. We demonstrate that the Tudor domain-based approach exhibits broad specificity for proteins harboring mono- or di-methylated arginines, encompassing both RGG/RG-rich and non-RG motifs,

facilitating the discovery of novel methylation sites. Using this strategy, we identify asymmetric dimethylarginine (aDMA) at the N-terminus of eIF3D, an essential component of the eukaryotic translation initiation complex. Biochemical analyses reveal that aDMA modification at R99 of eIF3D plays a regulatory role in protein translation initiation. Our findings establish a generally applicable approach for proteomic profiling of arginine methylation and unveil a novel regulatory role for this modification in eukaryotic protein translation.

Referee #1:

All of my concerns have been addressed.

Referee #2:

The authors have satisfactorily addressed my critique and I support publication of the manuscript.

Referee #3:

The authors did substantial amount of additional work upon the revision. The authors addressed my concerns and beyond while addressing the comments of the other two reviewers. I recommend publishing this work.

All editorial and formatting issues were resolved by the authors.

Prof. Yanqiu Yuan
Sun Yat-sen University
China

Dear Prof. Yuan,

I am very pleased to accept your manuscript for publication in the next available issue of EMBO reports. Thank you for your contribution to our journal.

Yours sincerely,
